# De novo synthesized polyunsaturated fatty acids operate as both host immunomodulators and nutrients for *Mycobacterium tuberculosis*

Thomas Laval[1,2], Laura Pedró-Cos[1,2], Wladimir Malaga[3], Laure Guenin-Macé[1], Alexandre Pawlik[4], Véronique Mayau[1], Hanane Yahia-Cherbal[2,5], Océane Delos[6,7], Wafa Frigui[4], Justine Bertrand-Michel[6,7], Christophe Guilhot[3], Caroline Demangel[1]*

[1]Immunobiology of Infection Unit, Institut Pasteur, INSERM U1224, Université de Paris, Paris, France; [2]Université de Paris, Sorbonne Paris Cité, Paris, France; [3]Institut de Pharmacologie et de Biologie Structurale (IPBS), Université de Toulouse, CNRS-UPS UMR 5089, Toulouse, France; [4]Integrated Mycobacterial Pathogenomics Unit, Institut Pasteur, CNRS UMR 3525, Université de Paris, Paris, France; [5]Immunoregulation Unit, Institut Pasteur, INSERM U122, Université de Paris, Paris, France; [6]MetaboHUB-MetaToul, National Infrastructure of Metabolomics and Fluxomics, Toulouse, France; [7]I2MC, Université de Toulouse, INSERM, Université Toulouse III - Paul Sabatier (UPS), Toulouse, France

*For correspondence:
demangel@pasteur.fr

Competing interest: The authors declare that no competing interests exist.

**Abstract** Successful control of *Mycobacterium tuberculosis* (Mtb) infection by macrophages relies on immunometabolic reprogramming, where the role of fatty acids (FAs) remains poorly understood. Recent studies unraveled Mtb's capacity to acquire saturated and monounsaturated FAs via the Mce1 importer. However, upon activation, macrophages produce polyunsaturated fatty acids (PUFAs), mammal-specific FAs mediating the generation of immunomodulatory eicosanoids. Here, we asked how Mtb modulates de novo synthesis of PUFAs in primary mouse macrophages and whether this benefits host or pathogen. Quantitative lipidomics revealed that Mtb infection selectively activates the biosynthesis of $\omega$6 PUFAs upstream of the eicosanoid precursor arachidonic acid (AA) via transcriptional activation of *Fads2*. Inhibiting FADS2 in infected macrophages impaired their inflammatory and antimicrobial responses but had no effect on Mtb growth in host cells nor mice. Using a click-chemistry approach, we found that Mtb efficiently imports $\omega$6 PUFAs via Mce1 in axenic culture, including AA. Further, Mtb preferentially internalized AA over all other FAs within infected macrophages by mechanisms partially depending on Mce1 and supporting intracellular persistence. Notably, IFNγ repressed de novo synthesis of AA by infected mouse macrophages and restricted AA import by intracellular Mtb. Together, these findings identify AA as a major FA substrate for intracellular Mtb, whose mobilization by innate immune responses is opportunistically hijacked by the pathogen and downregulated by IFNγ.

## Editor's evaluation

In this study, the authors highlight a role for de novo biosynthesis of Poly-unsaturated Fatty Acids and the consequence effect of these metabolites on the production of arachidonic acid. The increased bio-availability of arachidonic acid seemingly promotes mycobacterial growth whilst inhibition of arachidonic acid formation, and its resultant downstream eicosanoid products, affect macrophage function but somewhat surprisingly do not affect growth of *M. tuberculosis* in macrophages

or in mice. The uptake of the different classes of fatty acids in axenic culture as well as in macrophages is explored and the authors demonstrate that the Mce1 transporter is largely responsible for their uptake during in vitro growth but only plays a partial role in their uptake during growth of the pathogen in host cells. This work will be of interest to bacteriologists and those studying infectious diseases.

## Introduction

*Mycobacterium tuberculosis* (Mtb), the causative agent of human tuberculosis (TB), caused 1.6 million deaths in 2017, and it is estimated that 23% of the world's population has a latent TB infection. This success is due to Mtb evolving sophisticated strategies to survive intracellularly in macrophages, its preferred habitat, for long periods of time (*Bussi and Gutierrez, 2019*). In particular, Mtb's capacity to import and metabolize host-derived lipids, including fatty acids (FAs) and cholesterol, contributes to long-term persistence in vivo (*Nazarova et al., 2017*; *Nazarova et al., 2019*; *Pandey and Sassetti, 2008*). At the macrophage level, Mtb infection triggers the formation of lipid droplets (LDs) whose FA content was proposed to serve as a nutrient source for intracellular Mtb (*Daniel et al., 2011*; *Peyron et al., 2008*; *Singh et al., 2012*). However, this view was challenged by a recent study showing that the IFNγ cytokine promotes LD accumulation by Mtb-infected macrophages while impairing the pathogen's capacity to acquire host-derived FAs (*Knight et al., 2018*). Whether Mtb infection and IFNγ signaling differentially impact on subcellular localization and dynamic redistribution of host FAs is largely unknown.

Mtb was shown to import fluorescently labeled saturated and monounsaturated fatty acids (SFAs and MUFAs, respectively) via a dedicated protein machinery named Mce1, which is coordinated with Mce4-mediated import of cholesterol and plays an important role in Mtb lipid homeostasis (*Laval et al., 2021*; *Lee et al., 2013*; *Nazarova et al., 2017*; *Nazarova et al., 2019*; *Wilburn et al., 2018*). In addition to SFAs and MUFAs, mammalian cells produce the additional subset of polyunsaturated fatty acids (PUFAs), whose secondary metabolites shape macrophage effector functions (*Dennis and Norris, 2015*; *Mayer-Barber and Sher, 2015*). In particular, the catabolism of arachidonic acid (AA) by cyclooxygenases (COX) and lipoxygenases (LOX) yields prostaglandins and lipoxins/leukotrienes, products referred to as eicosanoids that are important signaling molecules modulating inflammation and apoptosis. Interestingly, Toll-like receptors (TLRs) induce signal-specific reprogramming of FA synthesis in macrophages, with differential impact on antibacterial immunity (*Hsieh et al., 2020*). How Mtb-driven stimulation of TLRs reprograms PUFA and eicosanoid biosynthesis by host macrophages, and whether this promotes anti-mycobacterial immune responses remained to be determined.

Here, we combined quantitative lipidomics with genetic ablation and pharmacological inhibition approaches to assess the importance of the PUFA biosynthetic pathway in innate control of Mtb infection by macrophages. While PUFA biosynthesis contributed to the generation of inflammatory and antimicrobial responses in infected macrophages, its stimulatory effect on macrophage effector functions was not matched by an enhanced capacity to restrict intracellular Mtb infection. This led us to propose that newly generated PUFAs may serve as FA sources for intracellular Mtb. Our in vitro and cellular assays using traceable alkyne-FAs supported this hypothesis by showing that Mtb efficiently internalizes $\omega 6$ PUFAs in axenic cultures, and preferentially the eicosanoid precursor AA within macrophages. They also indicated that in cellulo, Mce1 partially contributes to Mtb's uptake of AA and supports intracellular persistence of the pathogen at late stages of infection. Together, our findings reveal the pro-inflammatory function of the PUFA biosynthetic pathway during Mtb infection and identify AA as a major FA source for intracellular Mtb. They support the view that Mtb draws on intracellular free AA generated by infection.

## Results

### Mtb infection stimulates the biosynthesis of SFAs, MUFAs, and upstream PUFAs by host macrophages

Mtb's impact on host FA metabolism was investigated by infecting bone marrow-derived macrophages (BMDMs), extracting total and free cellular FAs, and quantifying each FA species by gas

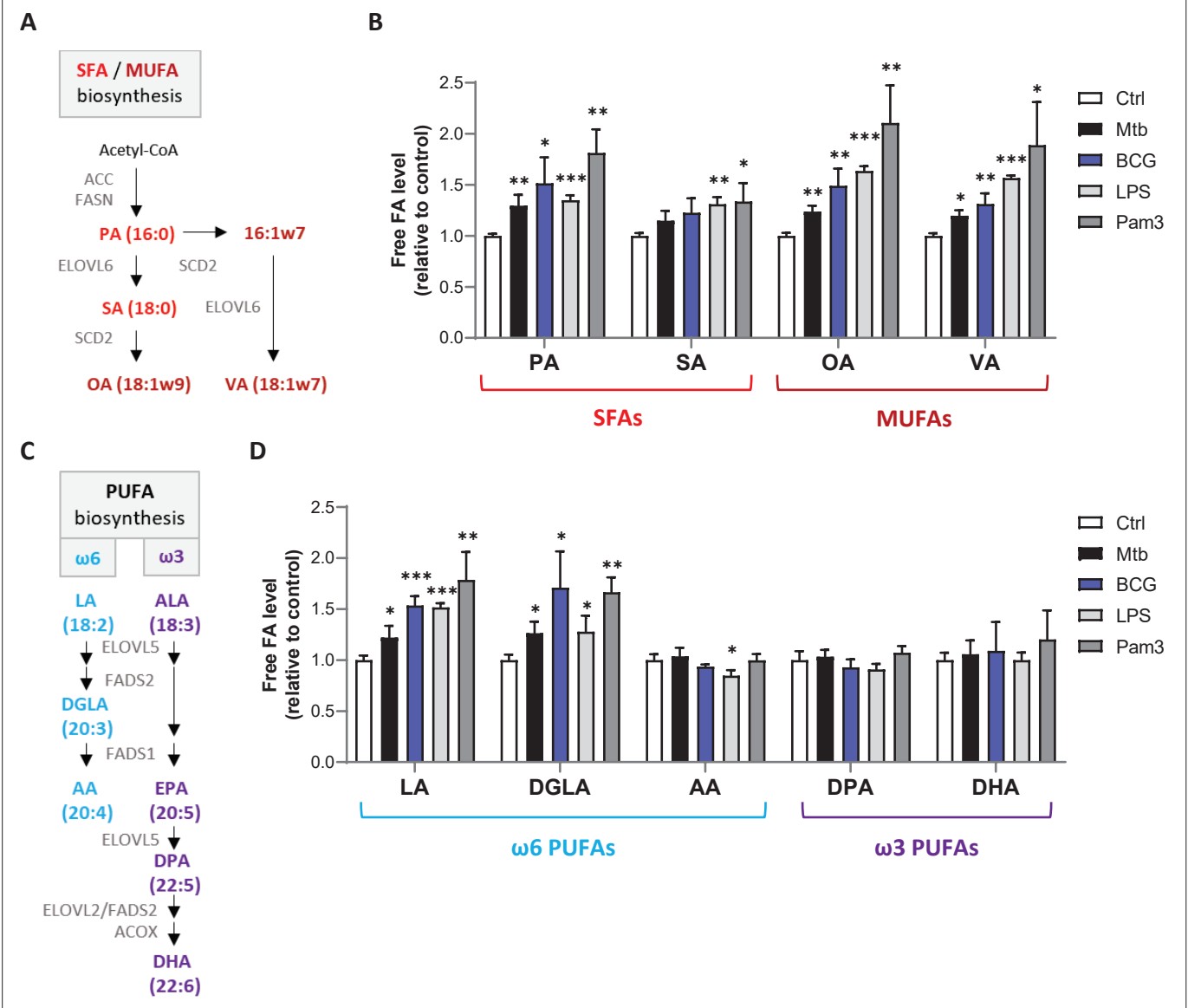

**Figure 1.** Mtb infection upregulates intracellular levels of free SFAs, MUFAs, and upstream PUFAs in host macrophages. (**A**) Schematics of SFA and MUFA biosynthetic pathways. PA, palmitic acid; SA, stearic acid; OA, oleic acid; VA, vaccenic acid. (**B**) Intracellular levels of free SFAs and MUFAs in BMDMs infected with *M. bovis* BCG (BCG) or *M. tuberculosis* H37Rv (Mtb) at the same multiplicity of infection (MOI) of 2:1, or treated with LPS or Pam3Csk4 (Pam3), or left untreated (Ctrl) for 24 hr. FA levels were normalized to total DNA content and are shown as fold change relative to Ctrl. (**C**) Schematics of PUFA biosynthetic pathways. LA, linoleic acid; DGLA, dihomo-γ-LA; AA, arachidonic acid; ALA, α-linolenic acid; DPA, docosapentaenoic acid; DHA, docosahexaenoic acid. (**D**) Intracellular levels of free PUFAs in BMDMs treated as in (**B**). All data are means ± SD (n = 3) and are representative of two independent experiments. *p<0.05, **p<0.01, ***p<0.001, unpaired Student's *t*-tests.

The online version of this article includes the following figure supplement(s) for figure 1:

**Figure supplement 1.** Mtb infection upregulates intracellular levels of total MUFAs and upstream PUFAs in host macrophages.

chromatography, with normalization to total DNA. Mtb triggered a significant increase in intracellular levels of free SFA palmitic acid (PA) and MUFAs (oleic acid [OA] and vaccenic acid [VA]) after 24 hr (*Figure 1A and B*). With regard to PUFAs, we detected an increased level of the ω6 precursor linoleic acid (LA) that was associated with elevated levels of its conversion product dihomo-gamma-linolenic acid (DGLA), but not the DGLA product AA (*Figure 1C and D*). On the ω3 PUFA side, α-linolenic acid (ALA) and eicosapentaenoic acid (EPA) were below detection limit, and the low levels of long-chain ω3 PUFAs docosapentaenoic acid (DPA) and docosahexaenoic acid (DHA) were not modulated by Mtb infection (*Figure 1C and D*). Infection-driven variations in free FA levels were associated with

parallel trends in total FA levels (*Figure 1—figure supplement 1*). These changes were not specific of the Mtb pathogen as BMDM infection with the *Mycobacterium bovis* BCG vaccine induced comparable FA profiles (*Figure 1B–D*, *Figure 1—figure supplement 1*), and they were no longer observed 48 hr post infection. When we stimulated BMDMs with TLR2/4 agonists, levels of free and total FAs were similarly modulated (*Figure 1B–D*, *Figure 1—figure supplement 1*), suggesting that Mtb-driven changes in FA levels in host macrophages result from recognition of mycobacteria pattern by TLR2/4.

De novo synthesis of SFAs is controlled by the FASN rate-limiting enzyme, and MUFAs are generated from SFAs by the SCD2 rate-limiting enzyme (*Figure 1A*; *Guillou et al., 2010*). In Mtb-infected BMDMs, increased levels of SFAs were associated with upregulation of *Fasn* transcript levels from 6 hr post infection (*Figure 2A*). *Scd2* mRNA expression was upregulated at 24 hr post infection with Mtb (*Figure 2A*), and the OA:SA ratio reflecting the efficacy of SFA conversion into MUFAs was increased upon Mtb and BCG infection (*Figure 2B*). Conversion of $\omega 6$ and $\omega 3$ PUFA precursors into downstream PUFAs is jointly controlled by three enzymes: the FA desaturases (FADS)1 and FADS2, and the elongase ELOVL5 (*Figure 1C*). *Fads1* and *Elovl5* transcript levels were transiently repressed at 6hr post infection with Mtb, while on the opposite those of *Fads2* were increased from 6hr until 24hr (*Figure 2C*). The inverse regulation of *Fads1*/*Elovl5,* compared to *Fasn*/*Fads2*, at 6hr post infection with Mtb was surprising since all genes are targets of the LXR and SREBP1 regulators of FA metabolism (*Daemen et al., 2013*; *Joseph et al., 2002*). It is interesting to note that the transient downregulation of *Fads1* and *Elovl5* gene expression correlated with a drop in LXR activity, reflected by decreased transcript levels of LXR target gene *Abca1* after 6 hr of Mtb infection, despite significant induction of *Nr1h3* gene expression. The sustained upregulation of *Fasn* and *Fads2* gene expression was associated with transcriptional induction of *Srebf1* and its target gene *Dhcr24* (*Figure 2D*). Together, data in *Figures 1 and 2* indicate that Mtb infection upregulates the biosynthesis of SFAs, MUFAs, and upstream PUFAs in host macrophages through activation of TLR2/4. They suggest that FA production results from activation of SREBP1, and that PUFA biosynthesis blockade downstream of FADS1 is due to a transient, post-transcriptional repression of LXR activity.

## IFNγ shuts down the biosynthesis of all FAs in Mtb-infected macrophages

Macrophage ability to mount efficient anti-Mtb responses relies on their activation by the Th1 cell-derived cytokine IFNγ, recently shown to limit host FA intake by intracellular Mtb (*Knight et al., 2018*). We thus sought to determine if IFNγ modulates FA biosynthesis by infected macrophages. Stimulating BMDMs with IFNγ prior to infection prevented Mtb-induced expression of *Fasn* and *Scd2* (*Figure 2A and C*), suggesting that IFNγ limits de novo synthesis of SFAs and MUFAs by infected macrophages. The effect of IFNγ on PUFA biosynthetic enzymes was more complex as the cytokine mitigated both the inhibitory effect of Mtb infection on *Elovl5* and *Fads1* gene expression after 6 hr, and its stimulatory effect on *Fads2* gene expression (*Figure 2C*). IFNγ-induced decrease in *Fasn* and *Fads2* expression correlated with reduced *Abca1* and *Dhcr24* transcript levels after 6 hr of infection (*Figure 2D*), suggesting that IFNγ prevents Mtb-induced stimulation of FA biosynthesis through downregulation of LXR and SREBP1 activity.

To assess the effect of such transcriptional changes on PUFA biosynthesis, we quantified macrophage's ability to convert a deuterated derivative of the $\omega 6$ precursor LA into downstream products upon infection with Mtb, with or without IFNγ priming, between 6 and 24 hr post infection. All $\omega 6$ intermediates (i.e., 20:2-d11, DGLA-d11, and AA-d11) were quantified, allowing us to measure the activity of each enzyme of the PUFA biosynthetic pathway via product to substrate ratios (*Figure 3A*). The measured percentages of each $\omega 6$ PUFA, relative to total deuterated FA, are also shown in *Figure 3—figure supplement 1A*. FADS2 activity was not modulated by Mtb in the conditions of the experiment, suggesting that infection-induced upregulation of *Fads2* expression (*Figure 2C*) takes more than 24 hr to translate into enhanced enzyme activity. In contrast, decreased *Fads1* expression at 6 hr post infection (*Figure 2C*) correlated with a marked reduction of FADS1-mediated conversion of DGLA in AA (*Figure 3A*), irrespective of IFNγ stimulation. Likewise, IFNγ-driven repression of *Fads2* gene expression in Mtb-infected BMDMs (*Figure 2C*) resulted in potent suppression of FADS2 activity (*Figure 3A*). Neither Mtb infection nor IFNγ stimulation affected the activity of ELOVL5. Therefore, the partial upregulation of the PUFA biosynthetic pathway that we observed in Mtb-infected

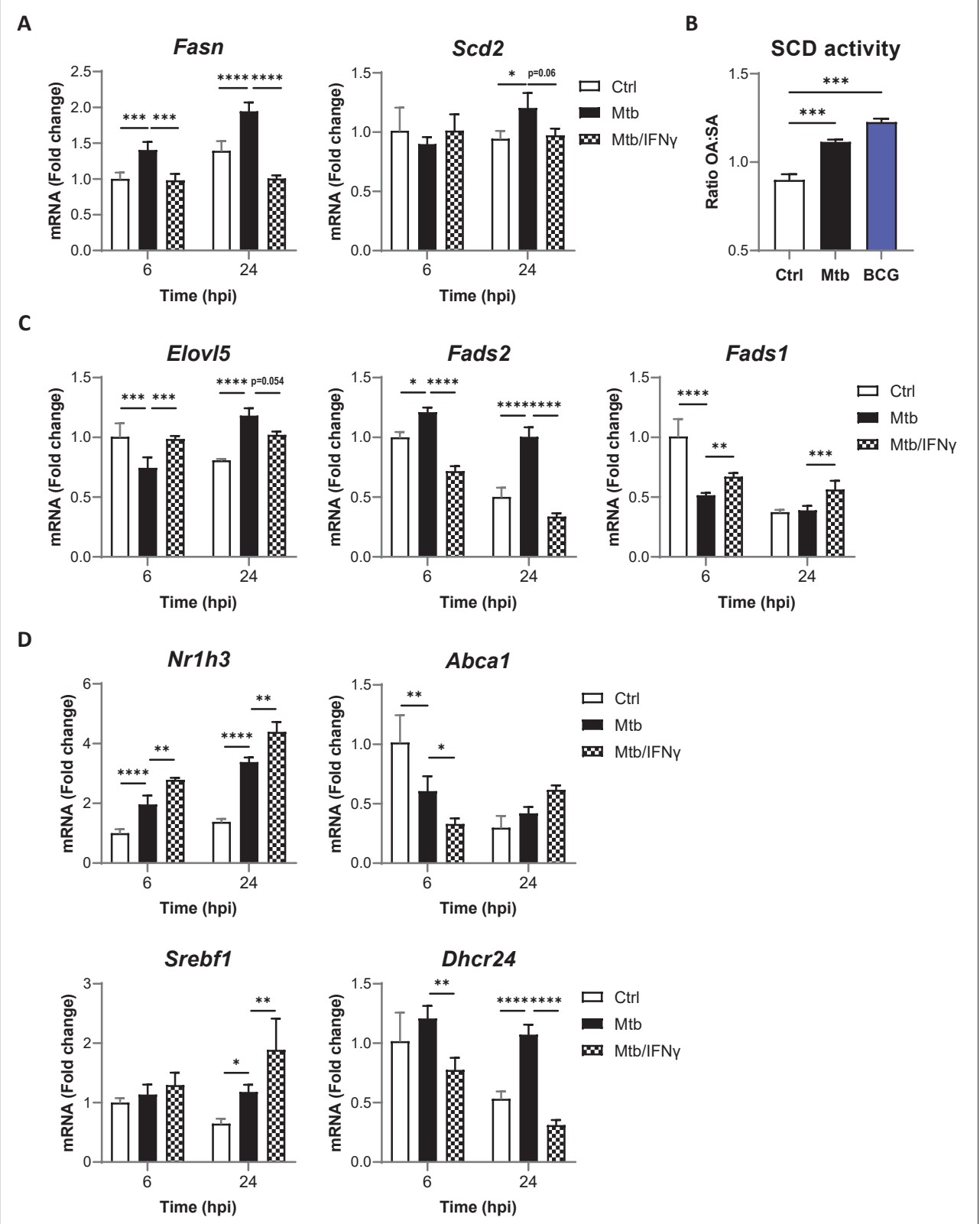

**Figure 2.** Mtb infection and IFNγ signaling cooperate to stop host PUFA biosynthesis. (**A**) Relative mRNA expression of SFA/MUFA biosynthetic enzymes in BMDMs primed with IFNγ before infection with Mtb for the indicated times, as determined by qRT-PCR. (**B**) SCD activity in BMDMs, as estimated by the ratio of oleic acid (OA) to stearic acid (SA) levels, after 24 hr of infection with Mtb or BCG. (**C–D**) Relative mRNA expression of biosynthetic enzymes (**C**) or LXR/SREBP1 target genes (**D**) in BMDMs treated as in (**A**), as determined by qRT-PCR. All data are means ± SD (n = 3) and

*Figure 2 continued on next page*

*Figure 2 continued*

are representative of two independent experiments. *p<0.05, **p<0.01, ***p<0.001, ****p<0.0001, two-way ANOVA with Dunnett post-hoc multiple comparison tests (**A**, **C**, **D**) and unpaired Student's *t*-tests (**B**).

macrophages was abrogated by cell exposure to IFNγ. In all, these results indicated that IFNγ shuts down the biosynthesis of all FAs in Mtb-infected macrophages.

## FADS2 inhibition impairs the effector functions of macrophages during Mtb infection

Long-chain PUFAs can be mobilized by hydrolysis of phospholipids to fuel the production of lipid mediators of inflammation (*Dennis and Norris, 2015*). In particular, conversion of AA by the COX/LOX pathways generates eicosanoids (*Figure 3B*), modulating the ability of macrophages to control myco-bacterial infection (*Mayer-Barber and Sher, 2015*). Although transcriptionally repressed, significant de novo synthesis of AA was maintained in Mtb-infected BMDMs (*Figure 3—figure supplement 1A*), suggesting a role in generation of eicosanoids. To test this, we used a selective inhibitor of FADS2 (SC-26196, hereafter named iFADS2) (*Obukowicz et al., 1999*). Exposing BMDMs to iFADS2 efficiently abrogated FADS2 activity in Mtb-infected, resting, and IFNγ-activated macrophages (*Figure 3A*), validating our experimental conditions.

Consistent with previous studies (*Chen et al., 2008*; *Knight et al., 2018*), BMDMs infected with Mtb upregulated *Ptgs2* expression (*Figure 3—figure supplement 1B*) and secreted higher amounts of AA metabolites deriving from both the COX (*Figure 3C*) and LOX pathways (*Figure 3D*) compared to noninfected controls. When BMDMs were infected with Mtb in the presence of iFADS2, the production of all COX/LOX-derived AA metabolites was significantly reduced (*Figure 3C and D*), suggesting that part of the infection-induced eicosanoids may originate from de novo synthesized PUFAs. Of note, upregulation of *Ptgs2* expression and production of PGE2 were both potentiated by BMDM exposure to IFNγ prior to infection with Mtb, and the inhibitory effect of iFADS2 on Mtb-driven production of PGE2 production was maintained in IFNγ-activated macrophages (*Figure 3—figure supplement 1B and C*).

PGE2 and LXA4 production by infected macrophages differentially influence the outcome of Mtb infection, promoting anti- or pro-mycobacterial responses via the induction of apoptotic or necrotic cell death, respectively (*Chen et al., 2008*; *Divangahi et al., 2010*; *Mayer-Barber and Sher, 2015*). Since iFADS2 treatment decreased Mtb-induced production of both PGE2 and LXA4 (*Figure 3C and D*), we tested how FADS2 inhibition affects the relative induction of apoptosis and necrosis in infected BMDMs (*Figure 3—figure supplement 1D*). Cell apoptosis and expression of *Syt7*, which is involved in lysosome-mediated membrane repair, were both downregulated by iFADS2 in Mtb-infected BMDMs, while necrosis levels remained unchanged (*Figure 3—figure supplement 1D and E*). We concluded that iFADS2-induced alterations of the PGE2/LXA4 balance results in a modest impairment of macrophage membrane repair and apoptosis during Mtb infection.

To determine if FADS2 inhibition alters the innate immune functions of macrophages, we profiled the expression of a panel of genes involved in antimicrobial and inflammatory responses using a custom NanoString nCounter CodeSet (*Supplementary file 1*). BMDMs were infected with Mtb and treated or not with iFADS2, and gene expression was assessed at 6 and 24 hr post infection. We detected a decrease in *Ptgs2* gene expression in iFADS2-treated BMDMs infected with Mtb, which may account for the observed decrease in PGE2 production (*Figure 3C*). Besides, FADS2 inhibition induced a significant downregulation of major inflammatory genes (*Tnf*, *Il1b*, *Il6*) (*Figure 3E*) and genes involved in antimicrobial responses of macrophages (*Nos2*, *Nox1*, *Irg1*, *Sod2*) (*Figure 3F*). Overall, our analyses of FADS2-inhibited macrophages indicated that the PUFA biosynthetic pathway promotes antimicrobial and inflammatory responses.

## Inhibiting FADS2 does not impact Mtb growth in vivo

Our data in *Figure 3* predicted that FADS2 inhibition should limit the macrophage capacity to restrict the intracellular growth of Mtb. To test this hypothesis, we infected resting or IFNγ-primed BMDMs with Mtb in the presence or absence of iFADS2 and monitored mycobacterial growth during 6 days by titrating colony-forming units (CFUs) in cell lysates. Pharmacological inhibition of FADS2 had no detectable effect on intracellular growth of Mtb (*Figure 4A*), neither in resting nor IFNγ-stimulated

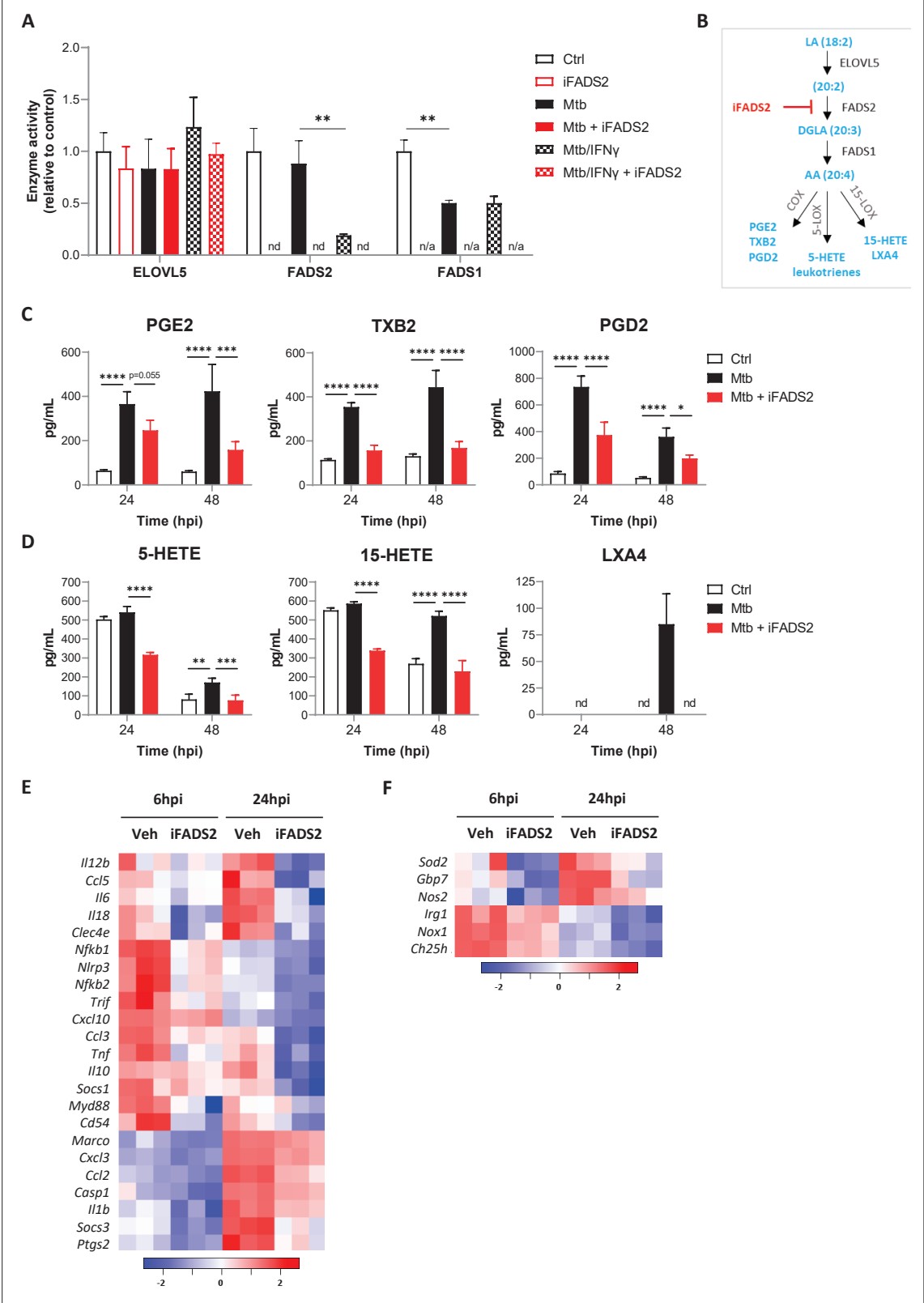

**Figure 3.** FADS2 inhibition impairs the effector functions of macrophages during Mtb infection. (**A**) Activities of PUFA biosynthetic enzymes in resting or IFNγ-primed BMDMs, either left untreated (Ctrl), or infected with Mtb and treated with a FADS2 inhibitor (iFADS2) or vehicle control, as determined by a conversion assay from 6 to 24 hr post infection using the ω6 precursor LA-d11. Enzyme activities were estimated with the ratio of deuterated fatty acid product to substrate levels and are shown as fold change relative to Ctrl. nd, product not detected; n/a, not applicable (substrate and product

*Figure 3 continued on next page*

*Figure 3 continued*

not detected). (**B**) Schematics of biosynthetic pathways of arachidonic acid (AA)-derived eicosanoids. COX, cyclooxygenase; LOX, lipoxygenase; PG, prostaglandin; TX, thromboxane; HETE, hydroxyeicosatetraenoic acid; LX, lipoxin. (**C, D**) Secreted levels of COX- (**C**) and LOX-derived (**D**) metabolites of AA by BMDMs either uninfected (Ctrl) or infected with Mtb, and treated with iFADS2 or vehicle control for 24 or 48 hr. Data in (**A**), (**C**), and (**D**) are means ± SD (n = 3), *p<0.05, **p<0.01, ***p<0.001, ****p<0.0001, unpaired Student's *t*-tests (**A**) and two-way ANOVA with Dunnett post-hoc multiple comparison tests (**C**, **D**). (**E, F**) Heatmap of mRNA expression levels of inflammatory (**E**) and antimicrobial (**F**) genes determined by NanoString analysis of BMDMs treated as in (**C**) for 6 or 24 hr. Shown are genes that were significantly downregulated by iFADS2 treatment at 6 and/or 24 hr (fold change of at least 1.15 and FDR < 0.05, two-way ANOVA with Benjamini–Hochberg adjustment for multiple comparison). Source data are available in *Figure 3— source data 1*.

The online version of this article includes the following source data and figure supplement(s) for figure 3:

**Source data 1.** Complete list of normalized mRNA levels in BMDMs, either noninfected (NI Ctrl) or infected with Mtb, and treated with iFADS2 (Mtb iFADS2) or vehicle control (Mtb Veh), as determined by NanoString analysis.

**Figure supplement 1.** Effects of IFNγ on PUFA biosynthesis by Mtb-infected BMDMs.

BMDMs. Similar results were obtained in BMDMs where *Fads2* expression was silenced by siRNA-mediated knock-down (*Figure 4—figure supplement 1A and B*). To determine if a complete defect in FADS2 expression would impact Mtb intracellular growth, we knocked out FADS2 in the THP-1 human cell line using the CRISPR-Cas9 approach. We selected three independent clones bearing distinct point mutations in exon 2 of *FADS2* gene and lacking FADS2 protein expression, and showed a near-complete defect in PUFA conversion (*Figure 4—figure supplement 1C and D*). In line with our data using iFADS2 and siRNAs, Mtb grew similarly in wild-type (WT) and knock-out (KO) clones (*Figure 4B*).

Since several lipid mediators and inflammatory genes modulated by iFADS2 are involved in adaptive immunity against Mtb (*Mayer-Barber and Sher, 2015*), we investigated the effect of a systemic inhibition of FADS2 in a mouse model of aerosol infection with Mtb. Mouse treatment with iFADS2 did not significantly alter Mtb growth in lungs and spleen in the conditions tested (*Figure 4C and D*). However, iFADS2 treatment altered the transcriptional induction of inflammatory (*Il1b, Il6, Tnf*) and antimicrobial (*Ptgs2, Nos2, Ch25h*) genes in lungs of Mtb-infected mice (*Figure 4E*). In line with our data obtained in macrophages, these results indicated that FADS2 promotes the generation of anti-mycobacterial responses. Blocking PUFA biosynthesis in infected hosts was nevertheless not sufficient to impair Mtb growth.

## Mtb efficiently imports ω6 PUFAs through the Mce1 transporter in axenic culture

We hypothesized that the anti-mycobacterial effects of PUFA biosynthesis on macrophage effector functions could be masked by a pro-mycobacterial role of PUFAs as nutrient sources for Mtb. Indeed, SFAs and MUFAs like PA and OA have been shown to be readily imported and metabolized by Mtb in axenic cultures and within macrophages (*Nazarova et al., 2017*; *Nazarova et al., 2019*). To determine if Mtb has the ability to import PUFAs, we used alkyne-FAs, structural analogs of natural FAs that can be detected by click-chemistry using azide-fluorochromes (*Thiele et al., 2012*; *Figure 5A*). Using this approach, we confirmed that PA, and OA to a lower extent, are efficiently internalized by Mtb (*Figure 5B*). While ω3 PUFAs (EPA and DHA) uptake was barely detected, we found that ω6 PUFAs (LA and AA) were imported by Mtb as efficiently as OA.

We next assessed the role of Mce1 as importer of PUFAs in Mtb by introducing inactivating mutations in genes of Mce operons by allelic exchange (*Figure 5C*, *Figure 5—figure supplement 1A*). In accordance with previous studies (*Nazarova et al., 2017*; *Nazarova et al., 2019*), knocking out *lucA*, *mce1D*, or *yrbE1A*, but not *yrbE2A*, *yrbE3A*, or *yrbE4A*, resulted in pronounced defects in alkyne-PA and -OA import (as shown for PA in *Figure 5D*). Likewise, we found that Mtb's import of alkyne-LA, -AA, -EPA, and -DHA was abrogated in the *ΔyrbE1A* (*Figure 5—figure supplement 1B*) and *Δmce1D* mutants (*Figure 5E*), and that FA import was restored in the *Δmce1D::comp* complemented strain (*Figure 5E*). This established Mtb's ability to import PUFAs via the Mce1 transporter.

Since ω6 PUFAs (LA and AA) were efficiently imported by Mtb via Mce1, we tested if they compete with other FAs for transport and measured Mtb's uptake of alkyne-FAs as above in the presence of increasing amounts of natural FAs. Uptake of alkyne-OA, -LA, and -AA was dose-dependently decreased by addition of their natural counterpart, validating the conditions of the assay (*Figure 5F*). Interestingly, LA and AA competed with each other and with OA, but not with PA (*Figure 5F*).

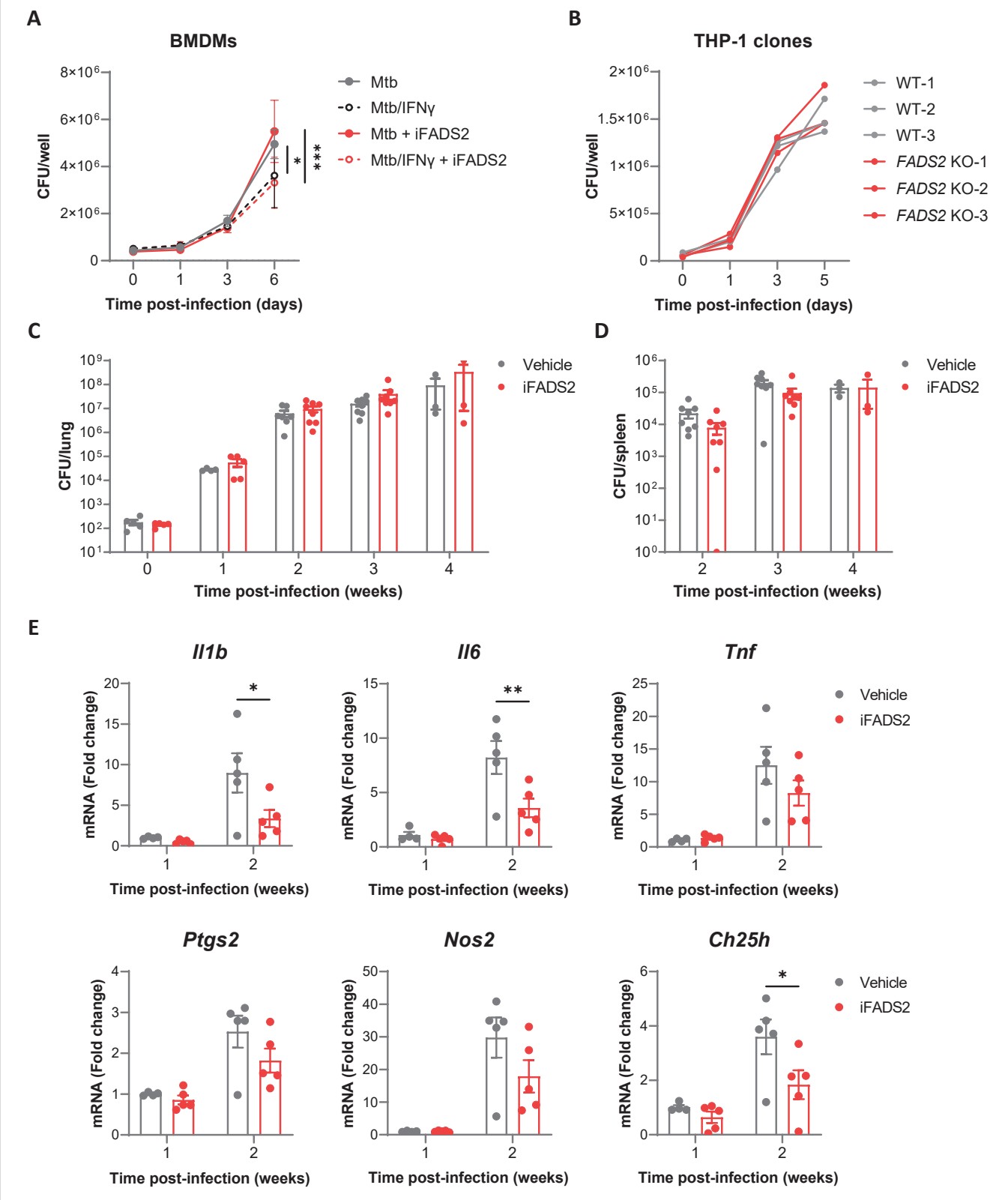

**Figure 4.** Inhibiting FADS2 does not impact Mtb growth in macrophages nor mice. (**A**) Intracellular growth of Mtb inside resting or IFNγ-primed BMDMs treated with iFADS2 or with vehicle control, as determined by colony-forming unit (CFU) plating at the indicated days post infection. Data are means ± SD (n = 3) and are representative of two independent experiments. *p<0.05, ***p<0.001, two-way ANOVA with Bonferroni post-hoc multiple comparison tests. (**B**) Intracellular growth of Mtb inside differentiated THP-1 wild-type (WT) or *FADS2* knockout (KO) clones, as determined by CFU

*Figure 4 continued on next page*

*Figure 4 continued*

plating at the indicated days post infection. Each line represents mean CFUs (n = 3) in one independent THP-1 clone. Data are representative of two independent experiments. (**C, D**) Growth of Mtb in the lungs (**C**) and spleen (**D**) of mice treated with iFADS2 or with vehicle control during 4 weeks, as determined by CFU plating. Data shown are means ± SEM of two pooled independent experiments (Exp 1: n = 3, Exp 2: n = 4–5 mice per time point). (**E**) Relative mRNA expression of inflammatory and antimicrobial genes in the lungs of mice treated as in (**C**), as determined by qRT-PCR. Data shown are means ± SEM (Exp 2: n = 4 or 5), *p<0.05, **p<0.01 in a two-way ANOVA with Bonferroni post-hoc multiple comparison tests.

The online version of this article includes the following source data and figure supplement(s) for figure 4:

**Figure supplement 1.** Limiting *Fads2* gene expression in macrophages.

**Figure supplement 1—source data 1.** Raw unedited gels.

**Figure supplement 1—source data 2.** Raw unedited gels.

**Figure supplement 1—source data 3.** Figures with uncropped gels.

Together, data in *Figure 5* indicated that SFAs, MUFAs, and ω6 PUFAs are imported by Mtb via Mce1 in axenic culture, with variable efficacy. They suggested that ω6 PUFAs compete with OA, but not PA, for uptake by Mce1.

## Mtb preferentially internalizes AA in the context of macrophages

We next investigated if all FAs had comparable ability to traffic to Mtb within infected macrophages. Here, BMDMs were infected with a GFP-expressing strain of Mtb prior to a pulse/chase with equivalent concentrations of alkyne derivatives of PA, OA, LA, AA, EPA, or DHA (*Figure 6—figure supplement 1A*). Intra-phagosomal Mtb was then isolated from infected cells, and FAs imported by Mtb were stained by click reaction and quantified by flow cytometry (*Figure 6—figure supplement 1B*). All alkyne-FAs could be detected in intracellular Mtb after 24 hr, and at lower levels after 72 hr (*Figure 6A*). Differently from in vitro grown Mtb (*Figure 5B*), intracellular Mtb showed a marked preference for AA over all other FAs including PA (*Figure 6A*). This observation was confirmed by confocal microscopy analysis of alkyne-AA signals in Mtb-infected cells (*Figure 6B*), and quantification of alkyne-PA and -AA signals in phagocytosed Mtb (*Figure 6C*).

Since IFNγ was previously shown to impair Mtb's import of Bodipy-PA during macrophage infection (*Knight et al., 2018*), we tested if the cytokine also reduces Mtb's uptake of other alkyne-FAs. After 24 hr of infection, Mtb's uptake of PA, OA, LA, and EPA was comparable in resting and IFNγ-activated BMDMs. However, bacterial import of AA was significantly decreased by IFNγ priming (*Figure 6—figure supplement 1C*). It is interesting to note that increased uptake of AA by intracellular Mtb correlated with a relatively higher internalization of this FA by host macrophages, irrespective of Mtb infection (*Figure 6D*). Similar differences in cellular and bacterial uptake of PA and AA were observed in human THP-1 macrophages (data not shown). This finding suggested that it is the intracellular bioavailability of AA that determines its import by Mtb, rather than intrinsic bacterial factors.

Finally, we assessed the role of Mce1 in PUFA uptake by intracellular Mtb using GFP-expressing versions of Mtb Δmce1D and Δmce1D::comp. In agreement with previous studies (*Nazarova et al., 2019*), Mtb's import of PA and OA was decreased by approximately 40% in the absence of Mce1D and restored by gene complementation (*Figure 6E*, *Figure 6—figure supplement 1D*). Mtb's import of LA and AA was also reduced by *mce1D* gene knockout, but to lower levels compared to PA and OA (*Figure 6E*, *Figure 6—figure supplement 1D*). To determine if bacterial import of FAs via Mce1D impacts the host immune response to infection, we compared the growth of Δmce1D, WT, and complemented Mtb strains in BMDMs. Although all strains displayed comparable intracellular growth during the first 3 days of BMDM infection (corresponding to the timeframe used in other experiments in this study), that of the Δmce1D mutant significantly decelerated after 6 days compared to WT and complemented strains (*Figure 6F*). We then asked whether this growth defect correlated with a differential eicosanoid profile by comparing the levels of AA metabolites produced by BMDMs infected with WT and Δmce1D Mtb (*Figure 6G*). Compared to WT Mtb, Δmce1D Mtb induced less COX-derived PGE2 and TXB2 and more LOX-derived 15-HETE and LXA4 (*Figure 6G*), an eicosanoid profile that is considered pro-mycobacterial. Furthermore, transcriptional induction of key inflammatory genes including *Pgts2* was significantly downregulated in macrophages infected with Δmce1D Mtb for 24 hr, compared to WT and complemented Mtb (*Figure 6—figure supplement 1E*). These findings are consistent with the defective pro-inflammatory responses of macrophages exposed to

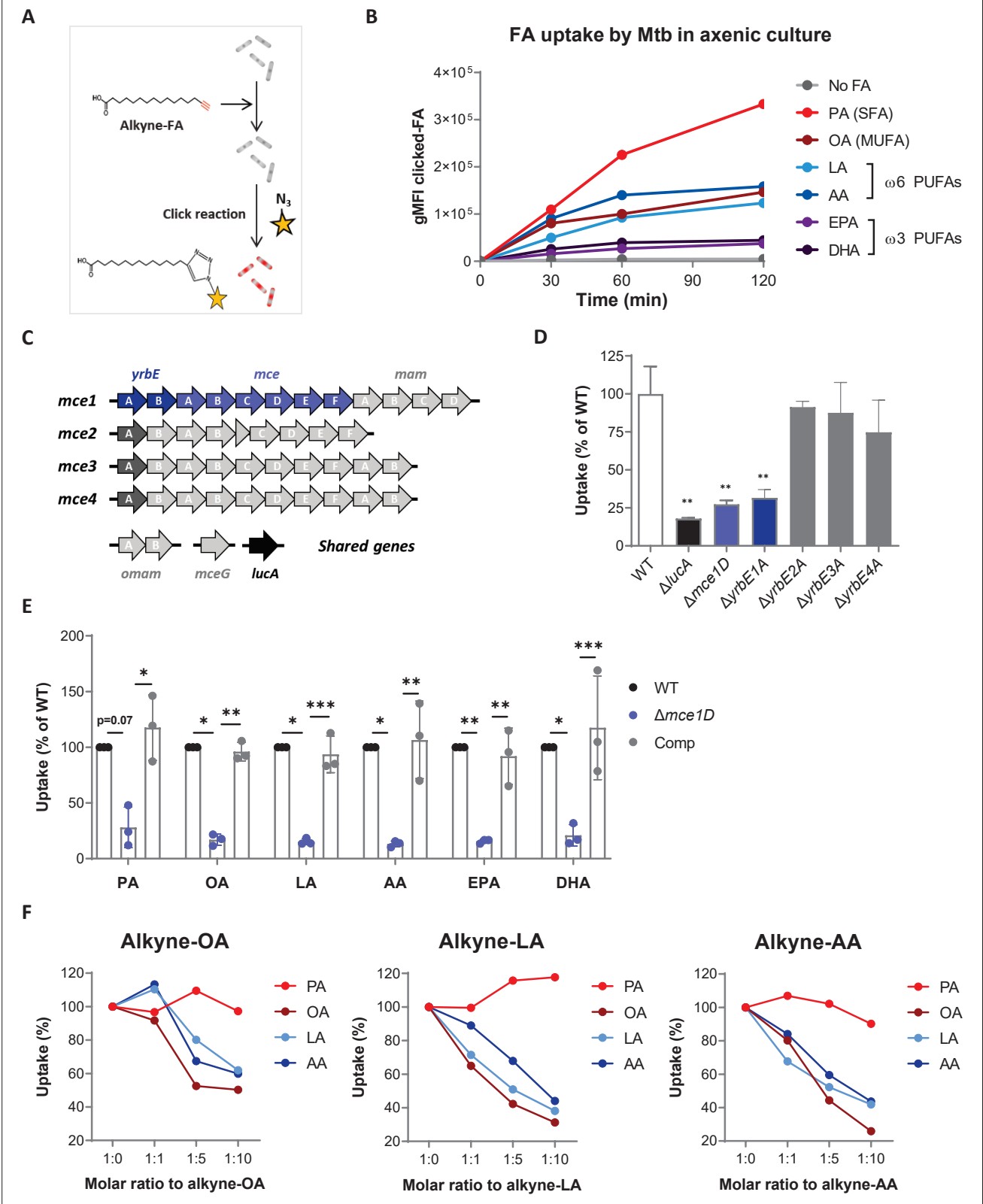

**Figure 5.** Mtb efficiently imports ω6 PUFAs through the Mce1 transporter in axenic culture. (**A**) Schematics of the click-chemistry approach used to compare the uptake of SFAs, MUFAs and PUFAs by Mtb in axenic culture. (**B**) Differential kinetics of alkyne-SFA, -MUFA, and -PUFA uptake (all added at a concentration of 20 μM) by Mtb, as estimated by flow cytometry assessment of the geometric mean fluorescence intensities (gMFI) of clicked-FA. Data shown are representative of three independent experiments. (**C**) Schematics of the composition of the *mce1-4* operons and related genes in Mtb

*Figure 5 continued on next page*

*Figure 5 continued*

genome. (**D**) Uptake of alkyne-PA by different Mtb strains deficient for the expression of genes belonging to *mce1-4* operons, relative to wild-type (WT) Mtb. Data are means ± SD (n = 3) and are representative of two independent experiments, **p<0.01, unpaired Student's *t*-tests. (**E**) Uptake of alkyne-FAs by Mtb Δ*mce1D* and its complemented counterpart (Comp), relative to WT. Data are means ± SD from three independent experiments, *p<0.05, **p<0.01, ***p<0.001, paired *t*-tests. (**F**) Relative uptake of alkyne-OA, -LA, and -AA by Mtb WT in the presence of increasing amounts of natural palmitic acid (PA), oleic acid (OA), linoleic acid (LA), or arachidonic acid (AA). Data shown are representative of at least two independent experiments.

The online version of this article includes the following source data and figure supplement(s) for figure 5:

**Figure supplement 1.** Inactivating genes of the Mce1 operon impairs Mtb's ability to import PUFAs.

**Figure supplement 1—source data 1.** Raw unedited gel.

**Figure supplement 1—source data 2.** Raw unedited gel.

cell wall lipids of an Mtb strain disrupted in the *mce1* operon (**Petrilli et al., 2020**). Together, our data in **Figure 6** revealed that in the context of macrophages Mtb preferentially imports AA over other host-derived FAs via mechanisms partially depending on the Mce1 transporter. They argue against eicosanoid production being limited by Mtb's capture of their AA precursor. Rather, they support the view that infection-induced mobilization of AA is opportunistically hijacked by Mtb to support intracellular growth.

## Discussion

In the present study, we show that $\omega$6 PUFAs, a host-specific subset of FAs, can be imported by Mtb via the Mce1 transporter. Our findings are consistent with previous observations that radiolabeled LA and ALA were incorporated into cell wall lipids of Mtb grown under axenic conditions (**Morbidoni et al., 2006**), and that AA used as a sole carbon source was able to support Mtb growth in vitro (**Forrellad et al., 2014**). $\omega$6 PUFAs were imported as efficiently as MUFAs and competed with MUFAs for Mce1-mediated import in Mtb. This suggests that MUFAs and $\omega$6 PUFAs share the same binding site on the Mce1 transporter or use a common accessory FA-binding protein that directs these FAs to the Mce1 transporter. Based on Mce1 gene inactivation studies (**Nazarova et al., 2019**; **Wilburn et al., 2018**), Mce1A-F proteins are believed to form a channel recognizing FA substrates and shuttling them across the mycobacterial cell wall to deliver them to the putative YrbE1A/B permeases embedded in the cytoplasmic membrane. Our observation that mutations in the *mce1D* and *yrbE1A* genes reduce Mtb's import of PUFAs by more than 80% in vitro supports the view that these proteins play a critical role in recognition and internalization of PUFAs, similar to SFAs and MUFAs. In comparison to all other FAs, $\omega$3 EPA and DHA were poorly internalized by Mtb in vitro. However, Mtb's uptake of $\omega$3 PUFAs also depended on Mce1 in axenic cultures. The distinctive conformational properties of natural, *cis* $\omega$3 PUFAs, such as high flexibility, may account for a lower affinity for the Mce1 receptor (**Chu and Wang, 2014**). Notably, OA, LA, and AA competed with each other for transport by Mce1, but none of these FAs did compete with PA. This suggests that Mce1 displays at least two distinct FA-binding sites specialized in recognition of SFAs and MUFAs/PUFAs, respectively, a property that may help Mtb optimize the uptake of host-derived FAs during infection.

Our lipidomic analyses demonstrated that Mtb infection triggers an increase in intracellular levels of free SFAs and MUFAs preceded by an upregulation of *Fasn* gene expression in host macrophages. We also observed that Mtb represses the expression of *Pparg* and several PPARγ gene targets involved in FA catabolism (*Cpt1a*, *Acox3*, *Lipa*) (**Figure 6—figure supplement 2**), which may further promote free SFA and MUFA accumulation in infected macrophages. Recent studies have identified the MyD88 signaling pathway as a major driver of SFA and MUFA metabolism reprogramming in TLR2/4-stimulated macrophages (**Hsieh et al., 2020**; **Oishi et al., 2017**). Cellular levels of SFAs and MUFAs were increased in BMDMs stimulated with TLR2 or TLR4 agonists, supporting the hypothesis that recognition of Mtb pattern by these receptors triggers SFA and MUFA biosynthesis. Likewise, total and free LA levels were upregulated by Mtb or TLR2 and TLR4 stimulation, suggesting an enhanced uptake of this essential $\omega$6 PUFA precursor by host macrophages upon infection.

While de novo synthesis of SFAs and MUFAs was globally upregulated by Mtb, the early PUFA biosynthetic pathway was stimulated through upregulation of *Fads2* gene expression but repressed at the level of FADS1. These findings are corroborated by a recent lipidomic study that detected a comparable blockade in long PUFA production by BMDMs stimulated with TLR2 or TLR4 agonists

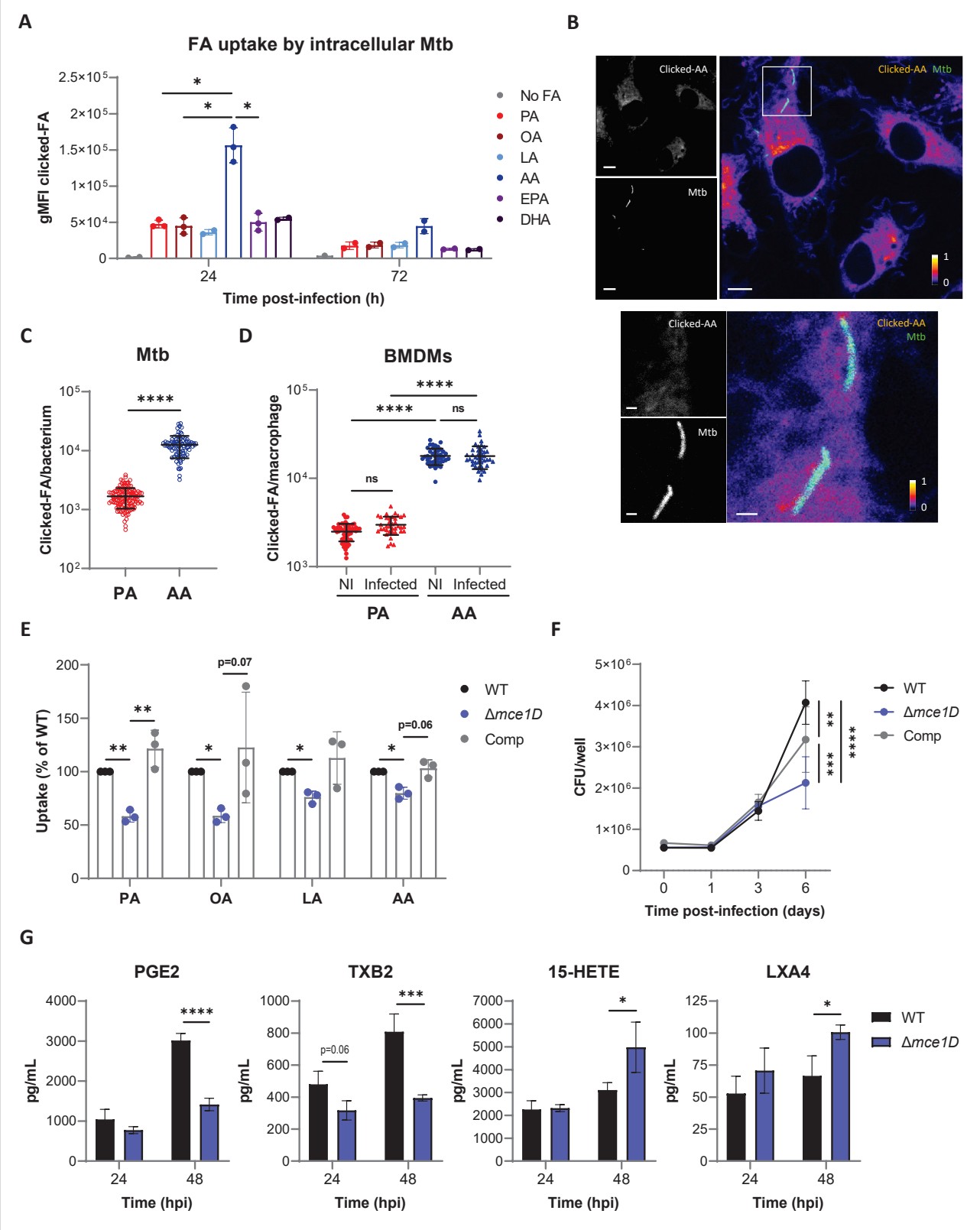

**Figure 6.** Mtb preferentially internalizes AA in the context of macrophages. (**A**) Differential uptake of alkyne-FAs by Mtb isolated from BMDMs infected for 24 or 72 hr, as measured by flow cytometry. Data are means ± SD from two or three independent experiments, *p<0.05, paired *t*-tests. (**B**) Distribution of alkyne-AA in Mtb-infected BMDMs at 24 hr post infection, as shown on representative confocal images (green = GFP-expressing Mtb, log scale colormap = clicked AA). Color bar indicates the relative range of pixel intensity (white = high, purple = low, from 0 arbitrary unit to 1). Bar scale

*Figure 6 continued on next page*

*Figure 6 continued*

= 5 µm. Enlargement of the boxed area in the merged image (bar scale = 1 µm). (**C, D**) Quantification of the alkyne-FA signal in intracellular bacteria detected based on GFP signal (**C**) and in whole BMDMs either noninfected (NI) or Mtb-infected (**D**) using confocal images of BMDMs infected for 24 hr with a GFP-expressing strain of Mtb. Bars show means ± SD, n > 82 for (**C**) and n > 37 for (**D**). ns, not significant, ****p<0.0001, unpaired Student's *t*-test (**C**) and one-way ANOVA with Tukey post-hoc multiple comparison tests (**D**). (**E**) Uptake of alkyne-FAs by GFP-expressing Mtb Δ*mce1D* and its complemented counterpart (Comp), recovered from BMDMs infected for 24 hr, relative to GFP-expressing Mtb WT, as analyzed by flow cytometry. Data are means ± SD from three independent experiments, *p<0.05, **p<0.01, paired *t*-tests. (**F**) Intracellular growth of different Mtb strains inside BMDMs, as determined by colony-forming unit (CFU) plating at the indicated days post infection. (**G**) Secreted levels of AA metabolites by BMDMs infected with Mtb WT or Δ*mce1D* for 24 or 48 hr. Data in (**F**) and (**G**) are means ± SD (n = 3), *p<0.05, **p<0.01, ***p<0.001, ****p<0.0001 in a two-way ANOVA with Bonferroni post-hoc multiple comparison tests.

The online version of this article includes the following figure supplement(s) for figure 6:

**Figure supplement 1.** Differential uptake of SFAs, MUFAs and PUFAs by intracellular Mtb and host BMDMs.

**Figure supplement 2.** Relative mRNA expression of PPARγ target genes in BMDMs infected with Mtb for the indicated times, as determined by NanoString analysis.

(*Hsieh et al., 2020*). Despite a decreased biosynthesis (*Figure 3A*), free and total AA levels were stable in Mtb-infected BMDMs (*Figure 1D*, *Figure 1—figure supplement 1*), implying that AA is mobilized from other sources to meet the cell's demands. Our studies of FA-pulsed BMDMs indicated that macrophages import exogenous AA with a particularly high efficacy compared to other FAs. Importantly, IFNγ priming of Mtb-infected macrophages fully blocked PUFA biosynthesis by downregulating *Fads2* gene expression. In addition, IFNγ selectively restricted AA import by intracellular Mtb, suggesting that IFNγ signaling activates several mechanisms preventing the pathogen from hijacking this FA.

An important new discovery was that Mtb imports AA at significantly higher levels, compared to all other tested FAs, in pulsed macrophages. In agreement with previous work by Nazarova et al. using macrophages pulsed with Bodipy-PA (*Nazarova et al., 2019*), *mce1D* contributed to about 40% of total PA uptake by intracellular Mtb (*Figure 6E*). In comparison, *mce1D* deficiency had a lower effect on the uptake of ω6 PUFAs, especially AA. Our observation that the differential uptake of PA and AA by intracellular Mtb correlates with the relative abundance of these FAs in the macrophage cytosol suggests that FA uptake by intracellular Mtb is primarily determined by their local bioavailability. Aside from Mce1, which transporters mediate host FA uptake by intracellular Mtb remains to be determined. Interestingly, intracellular persistence of the *mce1D* mutant of Mtb in macrophages was impaired at late stage of infection. This was not due to the generation of better anti-mycobacterial responses in host macrophages as compared to WT Mtb as the Δ*mce1D* mutant was a less good inducer of pro-inflammatory eicosanoids and cytokines in infected macrophages. Together with our observation that exogenous AA is superiorly imported by macrophages compared to other FAs, these findings support the view that Mce1-mediated import of AA plays an important role in Mtb survival within macrophages.

The AA pathway has recently emerged as a potential target of host-directed therapeutic strategies for TB. While the pro-inflammatory action of eicosanoids is beneficial to the host at early stages of infection by promoting the intracellular killing of Mtb via induction of TNFα in infected macrophages, it may drive immunopathology at later stages of disease (reviewed in *Young et al., 2020*). Nonsteroidal anti-inflammatory drugs inhibiting COX activity are being evaluated in humans as adjunctive therapies to standard TB treatments, with the rationale to limit inflammation-induced tissue pathology. In our study, transcriptional repression of *Fads1* in Mtb-infected macrophages did not prevent the production of AA-derived secondary metabolites, especially the pro-inflammatory COX products PGE2, PGD2, and TXB2 (*Figure 3*). Inhibiting FADS2 had a global anti-inflammatory effect during Mtb infection both in vitro and in vivo, which was associated with induction of a pro-mycobacterial eicosanoid profile in infected macrophages. Therefore, iFADS2 may increase the efficacy of COX inhibitors in immunopathological conditions, while depriving Mtb of additional FA sources.

## Materials and methods

### Reagents

All solvents for lipid extractions, antibiotics, Phorbol 12-myristate 13-acetate (PMA), fatty acid-free bovine serum albumin (FA-free BSA), ethyleneglycol- bis(β-aminoethyl)-N,N,N′,N′-tetraacetic acid (EGTA), methylcellulose (Methocel A4M), tyloxapol, Tween 20, and Triton X-100 were from Sigma-Aldrich. The FADS2 inhibitor SC-26196 (iFADS2) was stored as DMSO stock solutions at –20°C. All natural, deuterated, and alkyne derivatives of FAs were obtained from Cayman Chemical and stored as ethanol stock solutions at –20°C. Before medium supplementation, FAs were conjugated to albumin by a preincubation with FA-free BSA at a molar ratio of 2:1 (FA:albumin) 20 min at 37°C, unless otherwise stated.

### Macrophage generation and culture conditions

Bone marrow was isolated by perfusion of femurs and tibias from male 7- to 12-week-old C57BL/6J mice and cultured in Dulbecco's modified Eagle's medium (DMEM, Gibco Laboratories) with 10% heat-inactivated fetal bovine serum (FBS, PAA), 2 mM GlutaMAX, 15% L929-conditioned medium, 100 U/mL penicillin, and 100 μg/mL streptomycin (Pen-Strep) for 6 days, with media change on day 3. After differentiation, BMDMs were cultured in DMEM with 10% FBS, 2 mM GlutaMAX, and 5% L929-conditioned medium (BMDM media). For *Fads2* knockdown, BMDMs were transfected after 6 days of differentiation with 25 nM Fads2 siRNA pool (siFads2) or the same concentration of nontargeting siRNA pool (siCtrl) using lipofectamine RNAiMax according to the manufacturer's instructions. 24 hr post transfection, the medium was changed to BMDM media, and cells were used at 48 hr post transfection. The THP-1 cell line was purchased from and authenticated by ATCC, and was tested mycoplasma-negative before undertaking the reported experiments. THP-1 cells were cultured in RPMI-1640 (Gibco Laboratories) supplemented with 10% FBS, 2 mM GlutaMAX (THP-1 media) plus Pen-Strep. THP-1 monocytes were differentiated by incubation with 20 ng/mL PMA in THP-1 media for 48 hr, then PMA was removed and cells were maintained in THP-1 media for an additional 24 hr before infection.

### CRISPR/Cas9-mediated *FADS2* knockout in THP-1

A guide RNA targeting the exon 2 of *FADS2* gene was designed using http://crispor.tefor.net/ and cloned into the PX458 plasmid according to the published protocol (*Ran et al., 2013*). THP-1 cells were transfected with the plasmid using the Amaxa Nucleofector 2b device (Lonza) and the Human Monocyte Nucleofector Kit as previously described (*Schnoor et al., 2009*). The following day, GFP-positive transfected cells were sorted in 96-well plates at a density of 1 cell per well using a FACS ARIA III (BD Biosciences) and cultivated in THP-1 media. Growing cell clones were then screened by PCR using AmpliTaq master mix and primers to amplify the exon 2 of *FADS2* gene, and by FADS2 immunoblots as described below.

### Mycobacterial growth conditions

The H37Rv strain of Mtb and the Pasteur 1173P2 strain of *M. bovis* BCG and their DsRed-expressing, hygromycin-resistant counterparts were kindly provided by Laleh Majlessi (Institut Pasteur, Paris) and Nathalie Winter (INRA, Tours), respectively. Bacteria were routinely grown at 37°C in Middlebrook 7H9 broth (BD Biosciences) supplemented with 10% BD BBL Middlebrook OADC enrichment (BD Biosciences) and 0.05% tyloxapol. Kanamycin 40 μg/mL, hygromycin 50 μg/mL, streptomycin 25 μg/mL, and zeocin 50 μg/mL were used for selection.

### Mtb mutant construction and growth conditions

For each target gene, allelic exchange substrates (AES) were constructed by generating PCR fragments with primers X1-X2 and X3-X4, genomic DNA from Mtb H37Rv and enzyme PrimeSTAR GXL. Such primers allowed introduction of DraIII or Van91I restriction sites (in red and green, respectively, in *Supplementary file 2*) at both ends of the PCR fragment. In parallel, the kanamycin (km) resistance cassette was amplified from plasmid pET26b using primers k1 and k2, which also introduced a DraIII restriction site at both ends of the 900 bp km cassette. The three fragments were digested using DraIII restriction enzyme (Thermo Fisher Scientific), ligated for 30min using T4 DNA ligase, and

cloned with the CloneJET PCR Cloning Kit. The various AES were checked by Sanger sequencing. Each AES was then amplified by PCR on a 3 kb fragment using primers X5 and X6 and purified using QIAquick PCR Purification kit (QIAGEN). For gene *yrbE4A*, the occurrence of multiple DraIII and Van91I restriction sites in the targeted sequence led us to modify the cloning strategy. Three PCR fragments were generated using primers I1-I2, I3-I4, and ki1-ki2 and purified. Equivalent amounts of each fragment were mixed and a fusion PCR was generated using primers I5 and I6 to generate a 3 kb PCR product used as the AES to disrupt gene *yrbE4A*. Each AES was transformed by electroporation into a recombinant H73Rv strain expressing the recombineering system (*van Kessel and Hatfull, 2007*) from plasmid pJV53H. Transformants were selected on km-containing plates. Ten colonies were randomly picked and analyzed by PCR using primers X1 and kmR, X4, and kmF and X1 and X7. A clone giving the expected PCR profile (*Figure 6—figure supplement 1D*) was selected for further experimentation. The pJVH53 plasmid was cured by subculturing and isolating the selected mutant strains. The loss of pJVH53 plasmid was checked by patching the isolated mutants on hygromycin plate. One hygromycin-sensitive clone for each construct was retained for further characterization. For complementation of the *mce1D* mutant, we first transferred the Rv165 cosmid (*Brosch et al., 1998*), which contains the Mce1 region, into the *Escherichia coli* strain DY380. A zeocin resistance cassette was amplified by PCR using primers Z8 and Z9 and plasmid pMVZ621 as a substrate and inserted between the XbaI and Eco147i restriction sites of a mycobacterial integrative plasmid derived from pMV361 (*Stover et al., 1993*) to give pWM430. A fragment containing the *E. coli* replicon, the mycobacterial integrase, *attP* site, and the zeocin cassette was then amplified by PCR using primers M1 and M2 from plasmid pWM430 (3.4 kb fragment). The M1 and M2 primers exhibit a 100 nucleotides identity with genes *yrbE1A* and *rv0178*, respectively, and a 20 nucleotides identity with plasmid pWM430. The 3.4 kb amplification fragment was then transformed into DY380:Rv165 after induction of *E. coli* recombineering system as described (*Sharan et al., 2009*). Transformants were selected on LB agar containing zeocin (50 µg/mL). Recombinant plasmids in which a large Mtb fragment going from gene *yrbE1A* to gene *rv0178* was inserted by recombination into the zeocin containing plasmid were identified by PCR amplification using primers 78 and Z3 or Z1 and F7 and DreamTaq Green polymerase. A plasmid giving the expected PCR amplification profile was retained for further analysis and named pWM431. This plasmid was transferred by electroporation into the *mce1D* mutant, and transformants were selected on zeocin-containing plates. The parental and recombinant Mtb strains were made GFP-positive by transfer of a mycobacterial replicative plasmid pWM251, derived from pMIP12 (*Le Dantec et al., 2001*), which carries the streptomycin resistance cassette from pHP45Ω (*Prentki and Krisch, 1984*) and the *gfp* under the control of the pBlaF* promotor.

## FA uptake assays in axenic cultures

Alkyne-FA import was quantified by a modification of a previously described uptake assay with radio-labeled FAs (*Nazarova et al., 2017*). For FA uptake assays, bacteria were grown to an $OD_{600}$ of 0.4–0.8 in 7H9 medium supplemented with 0.5% FA-free BSA, 2 g/L dextrose, 0.85 g/L NaCl, and 0.05% tyloxapol (AD enrichment). Cultures were diluted to $OD_{600}$ of 0.1 in 7H9-AD medium and incubated with 5 µM (unless otherwise stated) of alkyne-FAs pre-conjugated to albumin. For competition assays, natural FAs (5–50 µM) and alkyne-FAs (5 µM) were simultaneously added to Mtb cultures. After an incubation of 1 hr (unless otherwise stated) at 37°C, bacterial cultures were collected by centrifugation, washed once in 7H9-AD medium and twice in ice-cold Wash Buffer (0.1% FA free-BSA and 0.1% Triton X-100 in PBS), and fixed in 4% PFA. Imported alkyne-FAs were then stained by a click reaction and analyzed by flow cytometry as described below for bacteria isolated from infected macrophages.

## Macrophage infections

For macrophage infection, mycobacteria were grown from frozen stocks to log phase at 37°C in 7H9-OADC. On the day of infection, bacteria were washed twice and resuspended in phosphate buffer saline, pH 7.4 (PBS). Bacteria were dissociated with the gentleMACS dissociator (Miltenyi Biotec) and remaining clumps were broken by passing through a syringe with a 25G needle and decanting the suspension for 10 min. Bacteria in suspension were quantified by spectrophotometry ($OD_{600}$) and diluted in cell culture medium. Where indicated, macrophages were primed with 20 ng/mL of IFNγ for 24 hr and pretreated with 0.1% DMSO as vehicle control or with 5 µM of the FADS2 inhibitor SC-26196 (iFADS2) for 2 hr prior to infection. Confluent macrophage monolayers were infected at

a multiplicity of infection (MOI) of 2 bacilli per cell (2:1) for 3 hr, extracellular bacteria removed by washing with warm DMEM or RPMI-1640 before adding BMDM or THP-1 media. For FADS2 inhibition experiments, the same concentration of iFADS2 was also added after phagocytosis. For enumeration of CFU, 2 volumes of water with 0.3% Triton X-100 were directly added to infected macrophages without supernatant removal. After 10 min at 37°C, lysed cells were serially diluted in water and plated on 7H11 agar supplemented with 0.5% glycerol and OADC. CFU were quantified after incubation at 37°C for 14–21 days.

## Apoptosis/necrosis assay

BMDMs were treated with iFADS2 and infected with Mtb WT as described above or treated with 5 µM of camptothecin (Sigma-Aldrich) as a positive control for apoptosis. At indicated times post infection, supernatants were harvested, cells were washed once in PBS, and incubated in TrypLE Express Enzyme 1X (Thermo Fisher Scientific) for 10 min at 37°C, 5% $CO_2$. Cells were harvested by flushing and pooled with corresponding supernatant and wash. After centrifugation for 4 min at 700 × g, harvested BMDMs were stained with DNA Nuclear Green DCS1 (Nucl) and Apopxin Deep Red (Apop) using the Abcam Apoptosis/necrosis Detection Kit following the manufacturer's instructions. Cells were then fixed in 4% PFA before FACS analysis of live (Apop$^-$ Nucl$^-$), apoptotic (Apop$^+$), and necrotic (Apop$^-$ Nucl$^+$) cells.

## Mouse studies

All animal experiments were performed in agreement with European and French guidelines (Directive 86/609/CEE and Decree 87-848 of 19 October 1987). The study received the approval by the Institut Pasteur Safety Committee (Protocol 11.245) and the ethical approval by the local ethical committee 'Comité d'Ethique en Experimentation Animale N° 89 (CETEA)' (CETEA 200037/ APAFiS #27688). Female, 7-week-old C57BL/6J mice were infected by aerosol route with Mtb H37Rv at a dose of 150–200 CFU per mouse with a homemade nebulizer as described (*Sayes et al., 2012*). 1 day prior to aerosol infection and 5 days a week thereafter, mice were given 100 mg/kg of FADS2 inhibitor SC-26196 by oral gavage. The lyophilized inhibitor was resuspended in 0.5% m/v methylcellulose – 0.025% Tween 20 as described (*Obukowicz et al., 1999*). At different time points, mice were sacrificed, and lungs and spleen were homogenized using a MM300 apparatus (QIAGEN) and 2.5 mm diameter glass beads. Serial dilutions in PBS were plated on 7H11 agar supplemented with 0.5% glycerol, 10% OADC, plus PANTA antibiotic mixture (BD Biosciences) for lung homogenates. CFUs were quantified after incubation at 37°C for 14–30 days.

## Confocal microscopy

BMDM monolayers cultured on glass coverslips in 24-well plates were infected with GFP-expressing strains of Mtb at a MOI of 2:1 for 3 hr, rinsed and incubated again in BMDM media. At 24 hr post infection, 5 µM of alkyne-FAs pre-conjugated to 1% FA-free BSA was added to the cells for 1 hr. This pulse was followed by a 1 hr chase in DMEM with 10% complete FBS. Infected macrophages were then washed in PBS, fixed in 4% PFA, and permeabilized in 0.5% Triton X-100. Internalized alkyne-FAs were then stained by a click reaction using the Click-iT Plus Alexa Fluor-647 Picolyl Azide kit according to the manufacturer's instructions. Nuclei were counterstained with DAPI, and coverslips were mounted onto glass slides with Prolong Diamond Antifade Mountant. Cells were imaged with an LSM 700 inverted confocal microscope and Zen Imaging software, and image analysis was performed using the Icy opensource platform (*de Chaumont et al., 2012*). For quantification of FA uptake, all acquisitions were performed using the same settings, and mean fluorescence intensity was measured in regions of interest corresponding to BMDMs or bacteria as defined with the HK-Means plugin of the Icy software.

## Flow cytometry quantification of FA uptake

Resting or IFNγ-primed BMDM monolayers were infected with GFP-expressing strains of Mtb at a MOI of 2:1 for 3 hr, rinsed and incubated again in BMDM media. At 1- or 3 days post infection, 5 µM of alkyne-FAs pre-conjugated to 1% FA-free BSA was added to the cells for 1 hr. This pulse was followed by a 1 hr chase in DMEM with 10% complete FBS. BMDMs were harvested, fixed in 4% PFA, and permeabilized with 0.5% Triton X-100 in PBS before staining by click reaction as described above and

FACS analysis. For the analysis of FA uptake by intra-phagosomal bacteria, these were isolated using a previously described procedure (*Nazarova et al., 2018*). After two washes in PBS-0.05% tyloxapol and one wash in Wash Buffer, isolated bacteria were fixed in 4% PFA. Fixed bacteria were then permeabilized by incubation with 0.25% Triton X-100 in PBS for 15 min at room temperature, and a click reaction was performed as described above. Stained bacteria were washed three times in ice-cold Wash Buffer and resuspended in 7H9 plus 0.85 g/L NaCl and 0.05% tyloxapol. Flow cytometry data were collected on a CytoFLEX flow cytometer (Beckman Coulter) and analyzed using FlowJo software.

## FA quantification by gas chromatography

BMDMs monolayers cultured in 100 mm cell culture dishes ($7–9 \times 10^6$ cells per dish) were infected as described above at a MOI of 2:1 with mycobacteria or treated with 100 ng/mL of LPS or Pam3Csk4. At indicated time points, cells were washed with ice-cold PBS (without $Mg^{2+}$ and $Ca^{2+}$) and harvested in ice-cold methanol/5 mM EGTA (2:1 v/v). For normalization, cellular DNA content was determined using the Quant-iT dsDNA Assay Kit as previously described (*Dennis et al., 2010*). Respectively 75%/25% of the sample was used for free/total FA extraction using heptadecanoic acid/glyceryl triheptadecanoate (Sigma-Aldrich) as internal standards. Lipids were extracted according to Bligh and Dyer (*Bligh and Dyer, 1959*) in dichloromethane/methanol/water (2.5:2.5:2, v/v/v), doped with 2 µg of suitable internal standard. After vortex and centrifugation at $2000 \times g$ for 10 min, the organic phase was dried under nitrogen and, for free FAs, directly trans-methylated with 1 mL of 14% boron trifluoride (BF3) in methanol and 1 mL of heptane for 5 min at room temperature. Total FAs were first hydrolyzed with 0.5 M potassium hydroxide (KOH) in methanol for 30 min at 50°C and trans-methylated in BF3-methanol plus heptane for 1 hr at 80°C. Methylated free FAs were extracted with hexane/water (3:1) as described previously (*Stuani et al., 2018*), dried and dissolved in ethyl acetate. 1 µL of methylated free or total FAs was analyzed by gas chromatography (GC) on a Clarus 600 Perkin Elmer system using FAMEWAX RESTEK fused silica capillary columns (30 m × 0.32 mm, 0.25 µm film thickness). Oven temperature was programmed from 110°C to 220°C at a rate of 2°C/min, and the carrier gas was hydrogen (0.5 bar). Injector and detector temperatures were set at 225 and 245°C, respectively. Peak detection and integration analysis were done using Azur software. FAs were quantified by measuring their area under the peak and normalizing to the internal standard. These quantities were then normalized by cellular DNA content.

## FADS2 activity assay

FADS2-mediated bioconversion of deuterated PUFAs was assessed as described previously (*Varin et al., 2015*). For BMDMs, resting or IFNγ-primed macrophages were treated with 5 µM of iFADS2 or vehicle control, and infected with Mtb as described above for 6 hr before addition of 4 µM LA-d11 pre-conjugated to FA-free BSA to the cell culture media. To check FADS2 knockout, THP-1 monocytes ($6.10^5$ cells/mL) were cultured in THP-1 media supplemented with 4 µM ALA-d14 pre-conjugated to FA-free BSA. In all cases, after an incubation of 18 hr (BMDM) or 24 hr (THP-1) with the deuterated PUFA, cells were washed twice in ice-cold PBS. BMDMs were harvested in ice-cold methanol/5 mM EGTA (2:1 v/v), while THP-1 were centrifuged and dry pellets were stored at –80°C. Lipids were extracted as described above, total FAs were hydrolyzed with KOH (0.5 M in methanol) at 50°C for 30 min, and derivatized for 20 min at room temperature with 1% pentafluorobenzyl-bromide and 1% diisopropylethylamine in acetonitrile as described (*Stuani et al., 2018*). Samples were dried, dissolved in ethyl acetate, and injected (1 µL) on a Thermo Fisher Trace GC system connected to a Thermo Fisher TSQ8000 triple quadrupole detector using a HP-5MS capillary column (30 m × 0.25 mm, 0.25 µm film thickness). Oven temperature was programmed as follows: 150°C for 1 min, 8°C/min to 350°C, then the temperature is kept constant for 2 min. The carrier gas was helium (0.8 mL/min). The injector, the transfer line, and the ion source temperature were set at 250, 330, and 300°C, respectively. Ionization was operated in negative ionization mode (methane at 1 mL/min) in selected ion monitoring (SIM) mode and 1 µL of sample was injected in splitless mode. The abundance of each fatty acid and their isotopologue containing 11 or 14 deuterium were obtained by integrating gas chromatographic signals using Trace Finder software.

## Lipid mediator profiling

For lipid inflammatory mediators analysis, resting or IFNγ-primed BMDMs monolayers cultured in 6-well plates were infected with Mtb at a MOI of 2:1. At indicated time points, cell supernatants were collected, methanol was added at a final concentration of 30%, and samples were transferred at –80°C. Samples were thawed, supplemented with 2 ng per sample of LxA4-d5, LTB4-d4, and 5-HETE-d8 (Cayman Chemical), and centrifuged at 5000 rpm for 15 min at 4°C. Cleared supernatants were submitted to solid-phase extraction using Oasis HLB 96-well clusters, and lipid inflammatory mediators were analyzed by LC/MS/MS as previously described (*Le Faouder et al., 2013*) on an Agilent LC1290 Infinity ultra-high-performance liquid chromatography system coupled to an Agilent 6460 triple quadrupole mass spectrometer (Agilent Technologies) equipped with electrospray ionization operating in the negative mode. Reverse-phase ultra-high-performance liquid chromatography was performed using a ZORBAX SB-C18 column (inner diameter: 2.1 mm; length: 50 mm; particle size: 1.8 µm; Agilent Technologies) with a gradient elution, and quantifications were obtained using Mass Hunter software.

## Immunoblot analyses

THP-1 monocytes were washed in cold PBS and lysed for 15 min in ice-cold lysis buffer (20 mM Tris, 150 mM NaCl, 1 mM EGTA, 1 mM MgCl$_2$, 1% n-dodecyl-(β)-d-maltoside, 50 mM sodium fluoride, and a protease inhibitor cocktail, all purchased from Sigma-Aldrich). Protein concentration was quantified with the NanoDrop-1000 Spectrophotometer (Thermo Fisher Scientific). Cell lysates were resolved on NuPAGE Bis-Tris gels and transferred to nitrocellulose membranes. Protein detections used the anti-FADS2 and anti-GAPDH antibodies. Protein complexes were revealed with the ECL Prime detection reagent (GE Healthcare) and chemiluminescence reading on a Fuji LAS-4000 Luminescent Image Analyzer.

## Gene expression analyses

At the indicated time points, infected macrophages were washed once in cold PBS, lysed with Qiazol lysis reagent, and total RNA was purified using miRNeasy kit and RNAse-free DNAse set according to the manufacturer's instructions (QIAGEN). RNA quality was checked using the Agilent RNA 6000 Nano Kit on the Agilent 2100 Bioanalyzer (Agilent Technologies). Gene expression was analyzed using the NanoString's nCounter digital barcode technology using a Custom CodeSet encompassing 112 genes including genes involved in metabolism and immune responses, and five housekeeping genes (*Supplementary file 1*). Reporter probes, sample, and capture probes were mixed together and hybridized at 65°C overnight. Following hybridization, samples were transferred and processed in the NanoString nCounter Prep Station. Data was collected by the NanoString nCounter Digital Analyzer, quality checked, and normalized to the housekeeping controls using nSolver analysis software according to NanoString analysis guidelines. For quantitative RT-PCR analysis, RNA quantity was assessed using the NanoDrop-1000 spectrophotometer and cDNA was prepared with the High-Capacity cDNA Reverse Transcription Kit. Quantitative PCR was performed using Power SYBR Green PCR Master Mix and gene-specific primers (*Supplementary file 3*). Amplification conditions on an Applied Biosystems StepOnePlus Real-time PCR system were 2 min at 50°C, 10 min at 95°C, followed by 40 cycles of 15 s at 95°C and 1 min at 60°C. Variations in expression were calculated by the $2^{-\Delta\Delta Ct}$ method using *Rpl19* as an endogenous control.

## Statistics

The GraphPad Prism 8 software was used for statistical comparisons and graphical representations, except for the heatmap, which was done using the nSolver analysis software. The statistical tests used are detailed in the figure legends.

## Acknowledgements

This study was supported by the Institut Pasteur (TL, CD) and INSERM (U1224, CD). We thank the Cytometry and Biomarkers (UTechS CB) Technological Platform for help with nCounter FLEX Nano-String use and the Image Analysis Hub of Institut Pasteur for confocal microscopy data analysis. We gratefully acknowledge the MetaToul (Toulouse metabolomics and fluxomics facilities, https://www.

metatoul.fr), part of the French National Infrastructure for Metabolomics and Fluxomics MetaboHUB-ANR-11-INBS-0010, for lipid analysis. We also thank Prof. Luke Chamberlain (University of Strathclyde, Glasgow, UK) for sharing his expertise of FA labeling by click-chemistry.

## Additional information

### Funding

| Funder | Grant reference number | Author |
| --- | --- | --- |
| Fondation pour la Recherche Médicale | 4th-year PhD scholarship from the FRM | Thomas Laval |
| Ministère de l'Enseignement supérieur, de la Recherche et de l'Innovation | 3 year PhD scholarship | Thomas Laval |
| Institut Pasteur | Core funding | Caroline Demangel |
| Institut Pasteur | 3 year doctoral scholarship, Pasteur-Paris University International Doctoral Program | Laura Pedró-Cos |

The funders had no role in study design, data collection and interpretation, or the decision to submit the work for publication.

### Author contributions

Thomas Laval, Conceptualization, Data curation, Formal analysis, Investigation, Methodology, Visualization, Writing - original draft; Laura Pedró-Cos, Wladimir Malaga, Véronique Mayau, Investigation; Laure Guenin-Macé, Formal analysis, Investigation, Methodology, Visualization; Alexandre Pawlik, Formal analysis, Investigation, Methodology; Hanane Yahia-Cherbal, Methodology; Océane Delos, Formal analysis, Investigation, Methodology, Océane performed some of the additional lipidomic assays requested by the reviewers. We confirm that all authors agree with her inclusion and place in the author list; Wafa Frigui, Investigation, Methodology, Supervision; Justine Bertrand-Michel, Conceptualization, Investigation, Methodology, Supervision, Validation, Visualization; Christophe Guilhot, Conceptualization, Funding acquisition, Investigation, Methodology, Project administration, Supervision, Validation, Visualization, Writing - original draft, Writing - review and editing; Caroline Demangel, Conceptualization, Formal analysis, Funding acquisition, Methodology, Project administration, Supervision, Validation, Visualization, Writing - original draft, Writing - review and editing

### Author ORCIDs

Thomas Laval (iD) http://orcid.org/0000-0002-9359-2783
Alexandre Pawlik (iD) http://orcid.org/0000-0002-5680-576X
Caroline Demangel (iD) http://orcid.org/0000-0001-7848-586X

### Ethics

All animal experiments were performed in agreement with European and French guidelines (Directive 86/609/CEE and Decree 87- 848 of 19 October 1987). The study received the approval by the Institut Pasteur Safety Committee (Protocol 11.245) and the ethical approval by the local ethical committee "Comité d'Ethique en Experimentation Animale N° 89 (CETEA)" (CETEA 200037 / APAFiS #27688).

### Decision letter and Author response

Decision letter https://doi.org/10.7554/eLife.71946.sa1
Author response https://doi.org/10.7554/eLife.71946.sa2

## Additional files

### Supplementary files

• Supplementary file 1. NanoString nCounter Custom CodeSet.

• Supplementary file 2. Primers used to generate the various allelic exchange substrates (AES) and *M. tuberculosis* (Mtb) mutants. DraIII and Van91I restriction sites introduced into the primers are indicated in red and green, respectively.

• Supplementary file 3. qPCR primers sequences.

• Transparent reporting form

### Data availability

All data analysed during this study are included in the manuscript and supporting files.

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

# Appendix 1

## Appendix 1—key resources table

| Reagent type (species) or resource | Designation | Source or reference | Identifiers | Additional information |
|---|---|---|---|---|
| Strain, strain background (*Mycobacterium tuberculosis*) | *M. tuberculosis* | Laleh Majlessi, Institut Pasteur, Paris | H37Rv | Animal passaged |
| Strain, strain background (*M. tuberculosis*) | *M. tuberculosis* DsRed | *Marino et al., 2015* | H37Rv DsRed | Animal passaged |
| Strain, strain background (*Mycobacterium bovis*) | BCG Pasteur | Roland Brosch, Institut Pasteur, Paris | 1173P2 | |
| Strain, strain background (*Mus musculus*), male | C57BL/6J | Charles River Laboratories | JAX stock no: 000664; RRID: IMSR_JAX:000664 | |
| Cell line (*Homo sapiens*) | THP-1 | ATCC | ATCC TIB-202; RRID:CVCL_0006 | |
| Strain, strain background (*Escherichia coli*) | HB101 | Promega | Cat# L2011 | Recombinant cloning and subcloning strain |
| Gene (*M. tuberculosis*) | *lucA* | ATCC27294 | *Rv3723* | |
| Gene (*M. tuberculosis*) | *mce1D* | ATCC27294 | *Rv0172* | |
| Gene (*M. tuberculosis*) | *mceG* | ATCC27294 | *Rv0655* | |
| Gene (*M. tuberculosis*) | *omamB* | ATCC27294 | *Rv0200* | |
| Gene (*M. tuberculosis*) | *yrbE1A* | ATCC27294 | *Rv0167* | |
| Gene (*M. tuberculosis*) | *yrbE2A* | ATCC27294 | *Rv0587* | |
| Gene (*M. tuberculosis*) | *yrbE3A* | ATCC27294 | *Rv1964* | |
| Gene (*M. tuberculosis*) | *yrbE4A* | ATCC27294 | *Rv3501* | |
| Genetic reagent (*M. tuberculosis*) | Δ*lucA* | This study, available from the corresponding author | Δ*lucA::km* | Chromosomal deletion of *Rv3723* (lucA) and insertion of an antibiotic resistance marker by double crossover recombination |
| Genetic reagent (*M. tuberculosis*) | Δ*mce1D* | This study, available from the corresponding author | Δ*mce1D::km* | Chromosomal deletion of *Rv0172* (mce1D) and insertion of an antibiotic resistance marker by double crossover recombination |
| Genetic reagent (*M. tuberculosis*) | Δ*mceG* | This study, available from the corresponding author | Δ*mceG::km* | Chromosomal deletion of *Rv0655* (mceG) and insertion of an antibiotic resistance marker by double crossover recombination |
| Genetic reagent (*M. tuberculosis*) | Δ*omamB* | This study, available from the corresponding author | Δ*omamB::km* | Chromosomal deletion of *Rv0200* (omamB) and insertion of an antibiotic resistance marker by double crossover recombination |
| Genetic reagent (*M. tuberculosis*) | Δ*yrbE1A* | This study, available from the corresponding author | Δ*yrbE1A::km* | Chromosomal deletion of *Rv0167* (yrbE1A) and insertion of an antibiotic resistance marker by double crossover recombination |

*Appendix 1 Continued on next page*

*Appendix 1 Continued*

| Reagent type (species) or resource | Designation | Source or reference | Identifiers | Additional information |
|---|---|---|---|---|
| Genetic reagent (*M. tuberculosis*) | *ΔyrbE2A* | This study, available from the corresponding author | *ΔyrbE2A::km* | Chromosomal deletion of *Rv0587* (*yrbE2A*) and insertion of an antibiotic resistance marker by double crossover recombination |
| Genetic reagent (*M. tuberculosis*) | *ΔyrbE3A* | This study, available from the corresponding author | *ΔyrbE3A::km* | Chromosomal deletion of *Rv1964* (*yrbE3A*) and insertion of an antibiotic resistance marker by double crossover recombination |
| Genetic reagent (*M. tuberculosis*) | *ΔyrbE4A* | This study, available from the corresponding author | *ΔyrbE4A::km* | Chromosomal deletion of *Rv3501* (*yrbE4A*) and insertion of an antibiotic resistance marker by double crossover recombination |
| Genetic reagent (*M. tuberculosis*) | *Δmce1D Comp* | This study, available from the corresponding author | *Δmce1D::km::yrbE1 to rv0178* | WT copy of the *yrbE1* to *rv0178* operon integrated at AttB |
| Recombinant DNA reagent | PX458 (plasmid) | Addgene (*Ran et al., 2013*) | pSpCas9(BB)–2A-GFP (pX458) | |
| Recombinant DNA reagent | pET26b (plasmid) | Sigma-Aldrich | pET26b(+) - Novagen | |
| Recombinant DNA reagent | pJV53H (plasmid) | This study, available from C. Guilhot | pJV53H | pJV53 (*van Kessel and Hatfull, 2007*) with hygromycin resistance gene |
| Recombinant DNA reagent | pMV361 (plasmid) | *Stover et al., 1993* | pMV361 | AttB integrating *M. tuberculosis* plasmid |
| Recombinant DNA reagent | pMVZ621 (plasmid) | Didier Zerbib, Toulouse Biotechnology Institute, France | pMVZ621 | pMV261 carrying a Zeocin resistance gene |
| Recombinant DNA reagent | pWM430 (plasmid) | This study | pWM430 | pMV361 with insertion of a zeocin cassette between the XbaI and Eco147i restriction sites |
| Recombinant DNA reagent | pWM431 (plasmid) | This study | pWM431 | pWM430 with a fragment going from gene *yrbE1A* to gene *rv0178* of *M. tuberculosis* H37Rv |
| Recombinant DNA reagent | pWM251 (plasmid) | This study | pWM251 | pMIP12 (*Le Dantec et al., 2001*) carrying the streptomycin resistance cassette from pHP45Ω (*Prentki and Krisch, 1984*) and the *gfp* under the control of the pBlaF* promotor |
| Recombinant DNA reagent | Rv165 (cosmid) | *Brosch et al., 1998* | Rv165 cosmid | Cosmid carrying a large fragment of the H37Rv genome covering the Mce1 region. |
| Sequence-based reagent | Primers used to generate AES and Mtb mutants | Merck | PCR primers | *Supplementary file 2* |
| Sequence-based reagent | | Eurofins Genomics | qPCR primers | *Supplementary file 3* |
| Sequence-based reagent | FADS2_ex2_F | Eurofins Genomics | PCR primer | 5'-GCACATTTCCAGTGCCAAGG-3' |
| Sequence-based reagent | FADS2_ex2_R | Eurofins Genomics | PCR primer | 5'-GGAGAGAGGAGACGCCACTA-3' |
| Sequence-based reagent | Guide RNA targeting the exon 2 of *FADS2* | Eurofins Genomics | Oligonucleotide | 5'-GCACCCTGACCTGGAATTCGT-3' |
| Transfected construct (*M. musculus*) | ON-TARGETplus siRNAs (SMARTpool) | Dharmacon/Horizon Discoveries | Non-targeting: D-001810-10 Mouse Fads2: L-049816-01 | |
| Antibody | Anti-FADS2 (rabbit polyclonal) | Thermo Fisher Scientific | PA576611; RRID:AB_2720338 | (1:1000) dilution |

*Appendix 1 Continued on next page*

*Appendix 1 Continued*

| Reagent type (species) or resource | Designation | Source or reference | Identifiers | Additional information |
|---|---|---|---|---|
| Antibody | Anti-GAPDH (rabbit monoclonal) | Cell Signaling Technology | 2118; RRID:AB_561053 | (1:1000) dilution |
| Antibody | Goat anti-rabbit IgG (HRP conjugate) | Santa Cruz | SC-2004; RRID:AB_631746 | (1:1000) dilution |
| Chemical compound, drug | SC-26196 | Cayman Chemical | 10792 | |
| Chemical compound, drug | Ultrapure LPS from *E. coli* | Enzo Life Sciences | ALX-581-013 | Serotype 055:B5 |
| Chemical compound, drug | Pam3Csk4 | Invivogen | tlrl-pms | |
| Chemical compound, drug | Lipofectamine RNAiMAX | Thermo Fisher Scientific | 13778075 | |
| Commercial assay or kit | VenorGeM Advance Mycoplasma detection kit | Minerva Biolabs | 11-7024 | |
| Commercial assay or kit | Human Monocyte Nucleofector Kit | Lonza | VPA-1007 | |
| Commercial assay or kit | Click-iT Plus Alexa Fluor-647 Picolyl Azide kit | Thermo Fisher Scientific | C10643 | |
| Commercial assay or kit | Quant-iT dsDNA Assay Kit, broad range | Thermo Fisher Scientific | Q33130 | |
| Commercial assay or kit | Apoptosis/ Necrosis Assay Kit | Abcam | ab176750 | |
| Commercial assay or kit | High-Capacity cDNA Reverse Transcription Kit | Thermo Fisher Scientific | 4368814 | |
| Commercial assay or kit | Power SYBR Green PCR Master Mix | Thermo Fisher Scientific | 4367659 | |
| Commercial assay or kit | miRNeasy Mini Kit | QIAGEN | 217004 | |
| Commercial assay or kit | CloneJET PCR Cloning Kit | Thermo Fisher Scientific | K1231 | |
| Commercial assay or kit | AmpliTaq Gold 360 master mix | Thermo Fisher Scientific | 4398876 | |
| Peptide, recombinant protein | DreamTaq Green polymerase | Thermo Fisher Scientific | EP0711 | |
| Peptide, recombinant protein | T4 DNA ligase | Thermo Fisher Scientific | EL0011 | |
| Peptide, recombinant protein | PrimeSTAR GXL DNA Polymerase | Takara Bio | R050B | |
| Software, algorithm | GraphPad Prism | GraphPad Prism (https://graphpad.com) | RRID:SCR_015807 | |
| Software, algorithm | Zen Imaging software | Zeiss | RRID:SCR_013672 | |
| Software, algorithm | Icy opensource platform | http://www.icy.bioimageanalysis.org *de Chaumont et al., 2012* | RRID:SCR_010587 | |

*Appendix 1 Continued on next page*

*Appendix 1 Continued*

| Reagent type (species) or resource | Designation | Source or reference | Identifiers | Additional information |
|---|---|---|---|---|
| Software, algorithm | FlowJo software | https://www.flowjo.com/ | RRID:SCR_008520 | |
| Software, algorithm | nSolver analysis software | http://www.nanostring.com/products/nSolver | RRID:SCR_003420 | |
| Other | 4–12% NuPAGE Bis-Tris gels | Thermo Fisher Scientific | NP0322BOX | |
| Other | iBlot 2 gel transfer Stacks Nitrocellulose system | Thermo Fisher Scientific | IB23001 | |
| Other | Prolong Diamond Antifade Mountant | Thermo Fisher Scientific | P36961 | |
| Other | DAPI stain | Sigma-Aldrich | D9542 | Used at 1 µg/mL |

