## [Editor Report]

In this study, the authors highlight a role for de novo biosynthesis of Poly-unsaturated Fatty Acids and the consequence effect of these metabolites on the production of arachidonic acid. The increased bio-availability of arachidonic acid seemingly promotes mycobacterial growth whilst inhibition of arachidonic acid formation, and its resultant downstream eicosanoid products, affect macrophage function but somewhat surprisingly do not affect growth of *M. tuberculosis* in macrophages or in mice. The uptake of the different classes of fatty acids in axenic culture as well as in macrophages is explored and the authors demonstrate that the Mce1 transporter is largely responsible for their uptake during in vitro growth but only plays a partial role in their uptake during growth of the pathogen in host cells. This work will be of interest to bacteriologists and those studying infectious diseases.

---

## [Decision Letter]

**Decision letter after peer review:**

Thank you for submitting your article "De novo synthesized polyunsaturated fatty acids operate as both host immunomodulators and nutrients for *Mycobacterium tuberculosis*" for consideration by *eLife*. Your article has been reviewed by 3 peer reviewers, and the evaluation has been overseen by Bavesh Kana as the Reviewing and Senior Editor. The following individuals involved in review of your submission have agreed to reveal their identity: Helena Boshoff (Reviewer #1); Frederick J Sheedy (Reviewer #2).

Essential revision:

1. It has been demonstrated that interferon-γ stimulation of macrophages drives lipid droplet formation and affects the availability of fatty acids derived from these lipid droplet to Mtb residing in the macrophage (doi: 10.1371/journal.ppat.1006874). Based on this, the authors should investigate if IFN-γ stimulation affects the biosynthesis of the fatty acids (for example PA and DGLA) in infected macrophages? Further, does IFN-γ stimulation affect acquisition by Mtb of newly synthesized fatty acids (eg. Arachidonic acid) from the host macrophage?

2. The lack of an effect with Fads2 inhibition on Mtb growth in macrophages and mice is perhaps surprising. It is the balance of eicosanoids (eg PGE2 versus LXA4 levels) that determines the outcome on macrophage membrane repair (and other effects in the context of an intact immune system in vivo – doi: 10.1038/ni.1758). An analysis of apoptosis versus necrosis in the BMDM model with or without FADS2 inhibition seems practically feasible to do and would provide important information regarding the effects of the FADS2 inhibitor on eicosanoid dysregulation in this system.

3. IFN-γ stimulation has a considerable effect on Cox2 expression and eicosanoid production in infected macrophages (doi: 10.4049/jimmunol.1600266). The authors should address the question of whether Fads2 inhibition in the context of IFN-γ stimulation affects Mtb survival in the macrophages?

4. A key experiment uses a specific inhibitor of Fads2 and monitors AA production from labelled exogenous Linoleic Acid. The finding that AA levels are decreased with increasing amounts of inhibitor are used to argue that this inhibitor is specific, but also that endogenous AA is generated from PUFA progenitors (Linoleic Acid and the intermediate, DGCA) during Mtb infection, despite a repression observed in the later desaturase enzyme, FADS1. However, to convincingly demonstrate this, this experiment should be repeated during Mtb infection. Although it is likely that control levels of AA may be low (given the role for AA import in bacterial growth), any differences with increasing amounts of inhibitor would support that this pathway is active during infection and can be targeted. An alternative approach would be to profile Fatty Acids in the same way as before, but after pre-treatment of Mtb-infected BMDM with the FADS2 inhibitor to show effects on Linoleic Acid, DGCA and AA production. This could be particularly interesting, or could be compounded by, the fact that FADS2 may also regulate the generation of Linoleic Acid from Oleic Acid to start with.

5. Striking effects are observed with the host response when FADS2 is targeted. Although various eicosanoid levels are decreased, which would argue that AA levels are reduced with inhibition, a profound global effect is observed on anti-microbial and inflammatory gene expression, including COX-2, an important upstream regulator of AA and eicosanoids. The observed effects on eicosanoid levels could be due to a broader effect on immune-responsive gene expression, which could be due to other "moon-lighting" roles for some of the PUFA pathway metabolites (LA, DGCA) or the enzymes (FADS2) themselves. Is this profound effect being phenocopied by the siRNA or CRISPR-deletion? This would strengthen the manuscript. What is the effect on eicosanoid production in these models and what happens the induction of inflammatory or anti-microbial genes?

6. The authors generate a novel bacterial strain in which import of AA is inhibited. However, the subsequent effect of this on bacterial growth in macrophages (by CFU plating) is not performed and could help delineate the various roles played by AA – as a bacterial nutrient or a host immunomodulatory signal. Side by side comparison of growth of the Mce1d mutant and wild-type Mtb strains in BMDM is required, as well as an examination of whether this has an effect on eicosanoid production and inflammatory/anti-microbial gene expression. This would clearly demonstrate competition between host and pathogen for AA, which could determine the outcome of the immune response.

*Reviewer #1 (Recommendations for the authors):*

Interferon-γ stimulation of macrophages drives lipid droplet formation and affects the availability of fatty acids derived from these lipid droplet to Mtb residing in the macrophage (doi: 10.1371/journal.ppat.1006874). Does IFN-γ stimulation affect the biosynthesis of the fatty acids (for example PA and DGLA) in infected macrophages? Does IFN-γ stimulation affect acquisition by Mtb of newly synthesized fatty acids (eg. Arachidonic acid) from the host macrophage?

The total lack of effect of Fads2 inhibition on Mtb growth in macrophages and mice is perhaps surprising. It is the balance of eicosanoids (eg PGE2 versus LXA4 levels) that determines the outcome on macrophage membrane repair (and other effects in the context of an intact immune system in vivo)( doi: 10.1038/ni.1758). An analysis of apoptosis versus necrosis in the BMDM model with or without FADS2 inhibition seems practically feasible to do and would provide important information regarding the effects of the FADS2 inhibitor on eicosanoid dysregulation in this system. Analysis of Syt-7 mRNA would also add information and possibly (but less important) LAMP1 expression.

IFN-γ stimulation has a considerable effect on Cox2 expression and eicosanoid production in infected macrophages (doi: 10.4049/jimmunol.1600266). Does Fads2 inhibition in the context of IFN-γ stimulation affect Mtb survival in the macrophages?

*Reviewer #2 (Recommendations for the authors):*

The evidence presented reveals an important role for the bioavailability of Arachidonic Acid in regulating both the host inflammatory response and mycobacterial growth. How these 2 pathways diverge needs to be examined further and can easily be demonstrated using some of the established tools available.

1) The authors examine the importance Fatty Acid Desaturase-2 (FADS2), a key enzyme in the PUFA metabolism pathway, which is upregulated after Mtb infection and promotes the generation of Arachidonic Acid from Linoleic Acid. A key experiment uses a specific inhibitor of this enzyme and monitors AA production from labelled exogenous Linoleic Acid. The finding that AA levels are decreased with increasing amounts of inhibitor are used to argue that this inhibitor is specific, but also that endogenous AA is generated from PUFA progenitors (Linoleic Acid and the intermediate, DGCA) during Mtb infection, despite a repression observed in the later desaturase enzyme, FADS1. However, to convincingly demonstrate this, this experiment should be repeated during Mtb infection. Although it is likely that control levels of AA may be low (given the role for AA import in bacterial growth), any differences with increasing amounts of inhibitor would support that this pathway is active during infection and can be targeted. An alternative approach would be to profile Fatty Acids in the same way as before, but after pre-treatment of Mtb-infected BMDM with the FADS2 inhibitor to show effects on Linoleic Acid, DGCA and AA production. This could be particularly interesting, or could be compounded by, the fact that FADS2 may also regulate the generation of Linoleic Acid from Oleic Acid to start with.

2) Secondly, striking effects are observed on the host response when FADS2 is targeted. Although various eicosanoid levels are decreased, which would argue that AA levels are reduced with inhibition, a profound global effect is observed on anti-microbial and inflammatory gene expression, including COX-2, an important upstream regulator of AA and eicosanoids. The observed effects on eicosanoid levels could be due to a broader effect on immune-responsive gene expression, which could be due to other "moon-lighting" roles for some of the PUFA pathway metabolites (LA, DGCA) or the enzymes (FADS2) themselves. If this profound effect is phenocopied by the siRNA or CRISPR-deletion approach would address this. What is the effect on eicosanoid production in these models and what happens the induction of inflammatory or anti-microbial genes?

3) Finally, a key finding is the elegant approach to delineate a role for PUFA and in particular, AA import by Mtb during macrophage infection. The authors generate a novel bacterial strain in which import of AA is inhibited. However, the subsequent effect of this on bacterial growth in macrophages (by CFU plating) is not performed and could help delineate the various roles played by AA – as a bacterial nutrient or a host immunomodulatory signal. Side by side comparison of growth of the Mce1d mutant and wild-type Mtb strains in BMDM is required, as well as an examination of whether this has an effect on eicosanoid production and inflammatory/anti-microbial gene expression. This would clearly demonstrate competition between host and pathogen for AA, which could determine the outcome of the immune response.

*Reviewer #3 (Recommendations for the authors):*

1. A major aspect of this study is the use of click chemistry to demonstrate uptake of AA by *M. tuberculosis* from the host during macrophage infection. However, it is well known that tuberculosis imports host fatty acids generally, and furthermore the fact that *M. tuberculosis* imports AA during macrophage infection was demonstrated in a previous publication (PMID: 10931146). Simply demonstrating uptake of specific FAs from the host does not advance our understanding of pathogenesis without mechanistic implications of this import being described.

2. One of the main conclusions of the manuscript is that AA is preferentially imported by *M. tuberculosis* relative to other FAs in the host. The authors own data demonstrate that labeled AA is taken up by macrophages more readily than palmitate (the only other FA evaluated), suggesting that the purported preference for AA by *M. tuberculosis* simply reflects abundance. The significance of this finding is unclear.

3. Treatment of mice and macrophages with an inhibitor of FADS2 showed no impact on the outcome of *M. tuberculosis* infection. The speculation that a pro-mycobacterial usage of PUFAs as nutrient sources by the bacteria masks the importance of AA synthesis by FADS2 for innate immunity is implausible and not substantiated by the data.

4. The authors claim that import of FAs by the bacteria equates to the utilization of the FA as a carbon source, however this is not necessarily that case. They would need to demonstrate metabolism of the FAs by the bacteria to make this claim.

5. The significance of the FA competition experiment is unclear and seems unrelated to the rest of the data in the manuscript.

6. In Figure 5, the requirement for Mce1 for uptake of AA from macrophages is different using the different assays of 5E (flow cytometry) and 5F (microscopy). Can the authors explain this discrepancy?

---

## [Author Response]

Essential revision:1. It has been demonstrated that interferon-γ stimulation of macrophages drives lipid droplet formation and affects the availability of fatty acids derived from these lipid droplet to Mtb residing in the macrophage (doi: 10.1371/journal.ppat.1006874). Based on this, the authors should investigate if IFN-γ stimulation affects the biosynthesis of the fatty acids (for example PA and DGLA) in infected macrophages? Further, does IFN-γ stimulation affect acquisition by Mtb of newly synthesized fatty acids (eg. Arachidonic acid) from the host macrophage?

As requested by the reviewer, we studied the effect of IFNγ stimulation on biosynthesis of FAs by Mtb-infected macrophages. New results are reported p.6, as follows:

“Macrophage ability to mount efficient anti-Mtb responses relies on their activation by the Th1 cell-derived cytokine IFNγ, recently shown to limit host FA intake by intracellular Mtb (Knight et al., 2018). We thus sought to determine if IFNγ modulates FA biosynthesis by infected macrophages. Stimulating BMDMs with IFNγ prior to infection prevented Mtb-induced expression of *Fasn* and *Scd2* (Figure 2A and 2C), suggesting that IFNγ limits de novo synthesis of SFAs and MUFAs by infected macrophages. The effect of IFNγ on PUFA biosynthetic enzymes was more complex, as the cytokine mitigated both the inhibitory effect of Mtb infection on *Elovl5* and *Fads1* gene expression after 6h, and its stimulatory effect on *Fads2* gene expression (Figure 2C). IFNγ-induced decrease in *Fasn* and *Fads2* expression correlated with reduced *Abca1* and *Dhcr24* transcript levels after 6h of infection (Figure 2D), suggesting that IFNγ prevents Mtb-induced stimulation of FA biosynthesis through downregulation of LXR and SREBP1 activity.

To assess the effect of such transcriptional changes on PUFA biosynthesis, we quantified macrophage’s ability to convert a deuterated derivative of the ω6 precursor LA into downstream products upon infection with Mtb, with or without IFNγ priming, between 6 and 24h post-infection. All ω6 intermediates (i.e. 20:2-d11, DGLA-d11 and AA-d11) were quantified, allowing us to measure the activity of each enzyme of the PUFA biosynthetic pathway via product to substrate ratios (Figure 3A). The measured percentages of each ω6 PUFA, relative to total deuterated FA, are also shown in Figure 3 —figure supplement 1A. FADS2 activity was not modulated by Mtb in the conditions of the experiment, suggesting that infection-induced upregulation of *Fads2* expression (Figure 2C) takes more than 24h to translate into enhanced enzyme activity. In contrast, decreased *Fads1* expression at 6h post-infection (Figure 2C) correlated with a marked reduction of FADS1-mediated conversion of DGLA in AA (Figure 3A), irrespective of IFNγ stimulation. Likewise, IFNγ-driven repression of *Fads2* gene expression in Mtb-infected BMDMs (Figure 2C) resulted in potent suppression of FADS2 activity (Figure 3A). Neither Mtb infection nor IFNγ stimulation affected the activity of ELOVL5. Therefore, the partial upregulation of the PUFA biosynthetic pathway that we observed in Mtb-infected macrophages is abrogated by cell exposure to IFNγ. In all, these results indicated that IFNγ shuts down the biosynthesis of all FAs in Mtb-infected macrophages.”

We also performed additional experiments to determine if IFNγ stimulation affects the acquisition of host FAs by Mtb (p.11):

“Since IFNγ was previously shown to impair Mtb’s import of Bodipy-PA during macrophage infection (Knight et al., 2018), we tested if the cytokine also reduces Mtb’s uptake of other alkyne-FAs. After 24h of infection, Mtb’s uptake of PA, OA, LA and EPA was comparable in resting and IFNγ-activated BMDMs. However, bacterial import of AA was significantly decreased by IFNγ priming (Figure 6 —figure supplement 1C).”

Discrepancy between our findings and those reported by Knight et al. could be due to the fact that we quantified FA uptake at 24h post-infection (because FA uptake was higher at this timepoint, as shown in Figure 6A), while Knight and colleagues examined PA uptake at 72h post-infection.

2. The lack of an effect with Fads2 inhibition on Mtb growth in macrophages and mice is perhaps surprising. It is the balance of eicosanoids (eg PGE2 versus LXA4 levels) that determines the outcome on macrophage membrane repair (and other effects in the context of an intact immune system in vivo – doi: 10.1038/ni.1758). An analysis of apoptosis versus necrosis in the BMDM model with or without FADS2 inhibition seems practically feasible to do and would provide important information regarding the effects of the FADS2 inhibitor on eicosanoid dysregulation in this system.

We thank the Reviewer for this pertinent suggestion. The requested analysis was performed, and our revised manuscript now includes new information regarding the effects of the FADS2 inhibitor on eicosanoid dysregulation (p.8):

“PGE2 and LXA4 production by infected macrophages differentially influence the outcome of Mtb infection, promoting anti- or pro-mycobacterial responses via the induction of apoptotic or necrotic cell death, respectively (Chen et al., 2008; Divangahi et al., 2010; Mayer-Barber and Sher, 2015). Since iFADS2 treatment decreased Mtb-induced production of both PGE2 and LXA4 (Figure 3C and 3D), we tested how FADS2 inhibition affects the relative induction of apoptosis and necrosis in infected BMDMs (Figure 3 —figure supplement 1D). Cell apoptosis and expression of *Syt7*, which is involved in lysosome-mediated membrane repair, were both significantly downregulated by iFADS2 in Mtb-infected BMDMs, while necrosis levels remained unchanged (Figure 3 —figure supplement 1D-E). We concluded that by downregulating PGE2, iFADS2 impairs macrophage membrane repair and apoptosis during Mtb infection. These data supported the view that FADS2 activity favors the generation of an anti-mycobacterial eicosanoid profile.”

3. IFN-γ stimulation has a considerable effect on Cox2 expression and eicosanoid production in infected macrophages (doi: 10.4049/jimmunol.1600266). The authors should address the question of whether Fads2 inhibition in the context of IFN-γ stimulation affects Mtb survival in the macrophages?

We confirmed Braverman et al.’ finding that IFNγ stimulation potently stimulates *Ptgs2* gene expression and PGE2 production (Figure 3 —figure supplement 1B-C). The inhibitory effect of iFADS2 on PGE2 production by Mtb-infected BMDMs was maintained in the context of IFNγ stimulation. However, similar to resting macrophages, FADS2 inhibition did not affect Mtb survival in IFNγ-primed macrophages (Figure 4A).

4. A key experiment uses a specific inhibitor of Fads2 and monitors AA production from labelled exogenous Linoleic Acid. The finding that AA levels are decreased with increasing amounts of inhibitor are used to argue that this inhibitor is specific, but also that endogenous AA is generated from PUFA progenitors (Linoleic Acid and the intermediate, DGCA) during Mtb infection, despite a repression observed in the later desaturase enzyme, FADS1. However, to convincingly demonstrate this, this experiment should be repeated during Mtb infection. Although it is likely that control levels of AA may be low (given the role for AA import in bacterial growth), any differences with increasing amounts of inhibitor would support that this pathway is active during infection and can be targeted.

A comparative analysis of de novo PUFA biosynthesis in uninfected and Mtb-infected BMDMs using deuterated LA-d11 is now provided (see also our answer to point #1). In line with our transcriptomic data shown in Figure 2, this analysis demonstrates that infection with Mtb induces a decrease in biosynthesis of AA-d11, that is aggravated by pre-exposure of the cells to IFNγ. Contrary to 20:2-d11, neither DGLA-d11 nor AA-d11 could be detected when BMDMs were treated with iFADS2, irrespective of Mtb infection or IFNγ stimulation (Figure 3 —figure supplement 1A). Together, these data establish that (i) endogenous AA is generated from PUFA precursors during Mtb infection in both resting and IFNγ-activated macrophages and (ii) iFADS2 efficiently blocks residual de novo AA biosynthesis in Mtb-infected macrophages.

An alternative approach would be to profile Fatty Acids in the same way as before, but after pre-treatment of Mtb-infected BMDM with the FADS2 inhibitor to show effects on Linoleic Acid, DGCA and AA production. This could be particularly interesting, or could be compounded by, the fact that FADS2 may also regulate the generation of Linoleic Acid from Oleic Acid to start with.

FA analysis did not reveal significant differences in FADS2 product levels between iFADS2-treated and control BMDMs (data not shown), indicating that PUFAs downstream of FADS2 can be mobilized from other sources. Being deficient in δ 12 desaturase, mammalian cells are unable to convert OA into LA, which is therefore essential and must be imported from the extracellular medium (Lee *et al.,* 2016). However, mammalian macrophages have been shown to produce mead acid (20:3ω9) from OA (18:1ω9) when they are deprived of the essential precursors of PUFAs (i.e. LA and ALA). Oleic to mead acid conversion involves the enzymes FADS1, FADS2 and ELOVL5 (Ichi *et al.,* 2014). The method used in our study to profile FA levels did not allow to specifically quantify mead acid. Thus, we cannot exclude that BMDMs produce mead acid in the conditions used, and one would expect that this biosynthetic pathway is impaired by iFADS2 as well.

5. Striking effects are observed with the host response when FADS2 is targeted. Although various eicosanoid levels are decreased, which would argue that AA levels are reduced with inhibition, a profound global effect is observed on anti-microbial and inflammatory gene expression, including COX-2, an important upstream regulator of AA and eicosanoids. The observed effects on eicosanoid levels could be due to a broader effect on immune-responsive gene expression, which could be due to other "moon-lighting" roles for some of the PUFA pathway metabolites (LA, DGCA) or the enzymes (FADS2) themselves.

We fully agree that repression of eicosanoid levels in iFADS2-treated cells could be due to other effects than just the decreased de novo synthesis of their AA precursor. As pointed out by the Reviewer, the decreased *Ptgs2* gene expression in iFADS2-treated BMDMs infected with Mtb may account for the observed decrease in PGE2 production. An additional sentence has been added in the relevant section of Results p.8 to acknowledge this possibility. It is interesting to note that BMDM production of 5-HETE was decreased by iFADS2 treatment after 24h of infection with Mtb (Figure 3D), despite the enhanced expression of *Alox5*, the gene encoding the 5-LOX enzyme that converts AA into 5-HETE (see Author response image 1). This result suggests that it is the lack of de novo synthesized AA that causes the drop in 5-LOX products in FADS2-inhibited macrophages (Figure 3D).

**Author response image 1. sa2fig1:** Relative mRNA expression of *Alox5* in Mtb-infected BMDMs treated as in Figure 3C, as determined by Nanostring analysis. Data are means +/- SD (n=3), *P<0.05, ****P<0.0001 in a two-way ANOVA with Bonferroni post-hoc multiple comparison tests.

Is this profound effect being phenocopied by the siRNA or CRISPR-deletion? This would strengthen the manuscript. What is the effect on eicosanoid production in these models and what happens the induction of inflammatory or anti-microbial genes?

To meet this demand, we compared the expression of a panel of genes amongst the most downregulated by iFADS2 treatment (Figure 3E) in BMDMs transduced with siRNA Fads2 or siRNA Ctrl for 48h prior to 6 to 24h of infection with Mtb. The left panel of the figure below shows that a significant drop in *Fads2* expression was achieved after 48h. However, the basal expression of *Il6* and *Ptgs2* was elevated in cells transduced with siFads2, compared to siCtrl (see Author response image 2, left), suggesting an immunometabolic adaptation to the lack of FADS2 activity. Infection-induced upregulation of *Il6* and *Ptgs2* was less important in BMDMs transfected with siFads2, compared to siCtrl (see Author response image 2, right). The other studied genes were not differentially induced by Mtb infection (data not shown). Therefore, the inhibitory effect of iFADS2 on inflammatory and anti-microbial genes was incompletely phenocopied by siRNA, potentially because siRNA-mediated silencing of FADS2 is only partial, and because compensatory pro-inflammatory mechanisms are implemented in FADS2-inhibited cells.

**Author response image 2. sa2fig2:** Relative mRNA expression of inflammatory genes in Mtb-infected BMDMs transfected with siCtrl or siFads2 for 48h (left panel), and infected with Mtb for the indicated periods of time (middle and right panels), as determined by qRT-PCR. Data are means +/- SD (n=3), *P<0.05, **P<0.01, ***P<0.001 in a two-way ANOVA with Bonferroni post-hoc multiple comparison tests.

6. The authors generate a novel bacterial strain in which import of AA is inhibited. However, the subsequent effect of this on bacterial growth in macrophages (by CFU plating) is not performed and could help delineate the various roles played by AA – as a bacterial nutrient or a host immunomodulatory signal. Side by side comparison of growth of the Mce1d mutant and wild-type Mtb strains in BMDM is required, as well as an examination of whether this has an effect on eicosanoid production and inflammatory/anti-microbial gene expression. This would clearly demonstrate competition between host and pathogen for AA, which could determine the outcome of the immune response.

The revised version of our manuscript now includes a side-by-side comparison of the growth of *Δmce1D*, WT and complemented Mtb strains in BMDMs (p.12):

“To determine if bacterial import of FAs via Mce1D impacts the host immune response to infection, we compared the growth of *Δmce1D*, WT and complemented Mtb strains in BMDMs. Although all strains displayed comparable intracellular growth during the first 3 days of BMDM infection (corresponding to the timeframe used in other experiments in this study), that of the *Δmce1D* mutant significantly decelerated after 6 days, compared to WT and complemented strains (Figure 6F). We then asked whether this growth defect correlated with a differential eicosanoid profile, by comparing the levels of AA metabolites produced by BMDMs infected with WT and *Δmce1D* Mtb (Figure 6G). Compared to WT Mtb, *Δmce1D* Mtb induced less COX-derived PGE2 and TXB2 and more LOX-derived 15-HETE and LXA4 (Figure 6G), an eicosanoid profile that is considered pro-mycobacterial. Furthermore, transcriptional induction of key inflammatory genes including *Pgts2* was significantly downregulated in macrophages infected with *Δmce1D* Mtb for 24h, compared to WT and complemented Mtb (Figure 6 —figure supplement 1E). These findings are consistent with the defective pro-inflammatory responses of macrophages exposed to cell wall lipids of an Mtb strain disrupted in the *mce1* operon (Petrilli et al., 2020). Together, our data in Figure 6 revealed that in the context of macrophages, Mtb preferentially imports AA over other host-derived FAs, via mechanisms partially depending on the Mce1 transporter. They argue against eicosanoid production being limited by Mtb’s capture of their AA precursor. Rather, they support the view that infection-induced mobilization of AA is opportunistically hijacked by Mtb to support intracellular growth.”